# Hydrological trade-offs due to different land covers and land uses in the Brazilian Cerrado

Jamil A.A. Anache[1], Edson Wendland[1], Lívia M.P. Rosalem[1], Cristian Youlton[1], Paulo T.S. Oliveira[2]

[1]Department of Hydraulics and Sanitation, São Carlos School of Engineering (EESC), University of São Paulo (USP), CxP. 359, São Carlos, SP, 13566-590, Brazil.
[2]Federal University of Mato Grosso do Sul, CxP. 549, Campo Grande, MS, 79070-900, Brazil.

*Correspondence to*: Jamil A.A. Anache (jamil.anache@usp.br)

**Abstract.** Farmland expansion in the Brazilian Cerrado, considered one of the largest agricultural frontiers in the world, has the potential to alter water fluxes on different spatial scales. Despite some large-scale studies being developed, there are still few investigations in experimental sites in this region. Here, we investigate the water balance components in experimental plots and the groundwater table fluctuation in different land covers: wooded Cerrado, sugarcane, pasture and bare soil. Furthermore, we identify possible water balance trade-offs due to the different land covers. This study was developed between 2012 and 2016 in the central region of the state of São Paulo, Southern Brazil. Hydrometeorological variables, groundwater table, surface runoff and other water balance components were monitored inside experimental plots containing different land covers; the datasets were analyzed using statistical parameters; and the water balance components uncertainties were computed. Replacing wooded Cerrado by pastureland and sugarcane shifts the overland flow (up to 42 mm yr$^{-1}$), and water balance residual (up to 504 mm yr$^{-1}$), and may affect groundwater table behavior. This fact suggests significant changes in the water partitioning in a transient land cover and land use (LCLU) system, as the evapotranspiration is lower (up to 719 mm yr$^{-1}$) in agricultural land covers than in the undisturbed Cerrado. We recommend long-term observations to continue the evaluations initiated in this study, mainly because tropical environments have few basic studies at the hillslope scale and more assessments are needed for a better understanding of the real field conditions. Such efforts should be made to reduce uncertainties, validate the water balance hypothesis and catch the variability of hydrological processes.

**Keywords:** Water balance, hillslope hydrology, sugarcane, pasture, Brazilian Cerrado.

## 1 Introduction

Brazil has significant areas used for extensive grazing over pasture, farmland (mainly soybeans and sugarcane), and part of these areas were mostly occupied by the native Cerrado, which decreased significantly in the last century (Marris, 2005). It is estimated that 52.2% of the original Cerrado area is now occupied by pasturelands and croplands, 0.8% by other land uses and 47% remains undisturbed (Beuchle et al., 2015). The conversion from native Cerrado to pastureland, and afterwards, to sugarcane, can be considered a potential land cover and land use change (LCLUC) in southeastern Brazil

(Alkimim et al., 2015). However, few studies investigate the effects of the LCLUC dynamics (Bonan, 2008; Loarie et al., 2011; Grecchi et al., 2014).

In the State of São Paulo, the minimum area required by the Brazilian forest code to maintain native forests in the Cerrado biome (20% of the total area of an estate) is not reached in much of the state (Soares-Filho et al., 2014). These areas were primary resources for expansion of the agricultural frontier in the tropics (Gibbs et al., 2010; Lapola et al., 2013), and these changes may cause significant disturbances in the hydrological processes (Loarie et al., 2011; Oliveira et al., 2014; Oliveira et al., 2015; Nobrega et al., 2017). These processes in tropical and subtropical zones are different from other regions across the world due to the increased solar energy availability and rainfall, calling for the need of basic field studies, long-term data acquisition and availability, and the development and application of mathematical models (Wohl et al., 2012; Burt and McDonnell, 2015).

In the context of the Cerrado biome, the conversion of undisturbed vegetation and pasturelands to mechanized crop systems (e.g. sugarcane, corn and soybeans) indicates that this region in Brazil has a dynamic LCLUC situation (Lapola et al., 2013). The sugarcane is the Brazilian backbone for energy security, as the ethanol production is the third most cultivated crop after soybeans and corn, reflecting the increasing demand for automotive fuels along the years (Leal et al., 2013; Rodrigues et al., 2018). Thus, the country is the world second largest ethanol producer and the Cerrado comprises the sugarcane expansion frontier due to the availability of water and pasturelands for the crop expansion (Bellezoni et al., 2018).

Native forests help to maintain the water cycle (Krishnaswamy et al., 2013; Ghimire et al., 2014). The evapotranspiration appears as a key component in the aquifer recharge control in the Guarani Aquifer System (GAS) outcrop zone (Wendland et al., 2007; Lucas and Wendland, 2015; Lucas et al., 2015). Thus, water partitioning in these areas should be well understood in order to have the responses from different land covers and land uses (LCLU), allowing the evaluation of possible water balance trade-offs due to the LCLUC (Krishnaswamy et al., 2013; Ghimire et al., 2014; Frank et al., 2014) caused by environmental and economic needs (Barretto et al., 2013).

In Brazil, many large-scale studies on water balance were developed in some hydrographic regions in the country (Paiva et al., 2013; da Paz et al., 2014; Oliveira et al., 2014; Melo et al., 2016). Nevertheless, experimental scale studies are still rare due to the local heterogeneities and uncertainties from hydrological measurements and estimates (Beven, 2006; Graham et al., 2010). Basic field-hydrology studies are important to improve the agricultural production efficiency while promoting sustainable development. Therefore, these studies are important to promote new solutions and techniques to maintain the water balance in spite of the rapid LCLUC (Dotterweich, 2013; Nobrega et al., 2017). Research of this kind can be done using experimental plots, which are delimitated hillslopes (control volume) where the overland flow is directed to one outlet (Sadeghi et al., 2013; Oliveira et al., 2015; Mwango et al., 2016; Oliveira et al., 2016; Strohmeier et al., 2016; Youlton et al., 2016b; Anache et al., 2017; Zhao et al., 2017; Anache et al., 2018).

The objective of this study is to evaluate the water fluxes in different LCLU: wooded Cerrado (also known as Cerrado *sensu stricto*), sugarcane, pasture and bare soil. In addition, we identify possible water balance trade-offs due to the different LCLU, compute the uncertainties for each water balance component and propagate the error separately for each LCLU.

## 2 Material and methods

### 2.1. Study sites and regional setting

The study sites are located in the Arruda Botelho Institute (IAB) in Itirapina, SP, Brazil (latitude 22º 11' 5" S, longitude 47º 51' 11" W, elevation 790 m a.m.s.l.) (Fig. 1). The area is inside two major hydrological hotspots in the country: the Guarani Aquifer System (GAS) (Wendland et al., 2007; Oliveira et al., 2017) outcrop zone and the Cerrado Biome. Site 1 is located under agricultural land covers (pasture, sugarcane and bare soil); while site 2 is located under the wooded Cerrado. The climate in the region is humid subtropical (Cwa, Köppen), with a hot and wet summer and a dry winter (Alvares et al., 2014). The average annual rainfall and temperature are 1486 mm yr$^{-1}$ and 21.5 °C, respectively (Cabrera et al., 2016). Additionally, the soil is sandy, it is classified as an Entisol (Oliveira et al., 2016), and its saturated hydraulic conductivity ranges from 11.30 mm h$^{-1}$ to 147.31 mm h$^{-1}$ along the superficial layer (first meter) (Oliveira, 2014; Youlton et al., 2016b).

**Insert Fig. 1**

### 2.2 Experimental setting and instrumentation

The experiment began in October, 2011 (Oliveira et al., 2015; Oliveira et al., 2016; Youlton et al., 2016b; Youlton et al., 2016a) and it is still in operation. This study considered the data collected during five years (2012-2016). It contains manual and automatic instruments, as well as permanent structures, all assembled in the two sites (Fig. 2). Two monitoring wells, meteorological stations (tripod and tower), and twelve steel-made bounded plots are the permanent structure. The instruments are described according to their positioning and function (Table 1). All instruments recorded data every 10 minutes, except the pressure transducers, which logged the groundwater table twice a day. The distance between sites 1 and 2 are 1.7 km. Site 1 has nine plots placed side by side (approximately 2.5 m of distance between each plot). Also in site 1, there is a meteorological station that concentrates almost all sensors placed at that site at one point, except for the soil moisture probes. They are connected to the meteorological station, but they are placed inside the plots (there is one placed inside the first sugarcane plot, and 20 m to the left, there is another one placed inside the first pasture plot). Site 2 has three plots inside a tropical woodland (wooded Cerrado) and due to the tree density and topography, the plots are approximately 5 to 10 m distant from each other. Approximately 50 m to the north from the plots, there is a meteorological tower (11 m height) containing all the sensors placed at site 2.

**Insert Fig. 2**

**Insert Table 1**

The plots were designed to adequately represent the process heterogeneities (Sadeghi et al., 2013): three replicates for each LCLU, 5 m width, 20 m length, and 9% slope gradient. The plots located at site 1 contain three different LCLU: (i) Pasture (*Brachiaria decumbens*) established 20 years ago, used for grazing and plant heights varying between 5 and 30 cm. The animals' (cattle) rotation period is 30 days (10 animals per hectare) and each one weighs approximately 420 kg. The animals remain in the area for 5 days in each rotation; (ii) contoured planted sugarcane (*Saccharum officinarum*) on beds

spaced 1.5 m apart. The plantation was established in October 2011, the canopy reaches at least 2 m high, and it was harvested every November; (iii) bare soil plots were maintained without plant cover by manual tillage and glyphosate application. The plots located at site 2 have the same experimental design as site 1 and contain (iv) wooded Cerrado as vegetation. The Cerrado comprises tropical vegetation in which the trees do not form a continuous canopy, however, it presents woody components of six to 7 m high (Alberton et al., 2014). The Cerrado vegetation is fire-resistant, it is considered a biodiversity hotspot and supports long dry periods (Brannstrom et al., 2008). Concerning the woodland characteristics, the wooded Cerrado area used in this study has 15522 individuals per hectare, the height of most of the trees is about 8 m, and the trees' diameters at breast height (DBH) are predominantly between 3 and 7 cm (Reys et al., 2013). The soil root zone in the wooded Cerrado may reach up to 18 m (Rawitscher, 1948). However, most of the water used for plants' transpiration comes from the first layers (up to 7.5 m) (Canadell et al., 1996; Oliveira et al., 2005; Garcia-Montiel et al., 2008).

## 2.3 Water balance components

The water balance components (Eq. 1) were monitored over 5 years (2012-2016) in a control volume defined by the bounded plots placed within the experimental sites. We considered different techniques to monitor these components according to the LCLU conditions. The water balance residual ($dS/dt$) includes subsurface flow, soil water storage, deep percolation, and groundwater recharge.

$$\frac{dS}{dt} = P - O_{\mathrm{F}} - E_{\mathrm{T}} \tag{1}$$

Where: $P$ is the rainfall (mm); $O_{\mathrm{F}}$ is the surface runoff (mm); $E_{\mathrm{T}}$ is the evapotranspiration (mm); and $dS/dt$ includes the soil water storage, subsurface flow and deep percolation (mm) during time $t$ (day, month or year).

The rainfall was monitored using a tipping bucket (Model TB4-L, Hydrological Services) with a gauging resolution of 0.254 mm (Table 1). The rainfall data was registered every 10 minutes, in order to obtain rainfall intensity and duration.

Rectangular experimental plots directed the overland flow to tanks at the end of the slope, where the volume was measured at times after rainfall events. We also calculated the runoff coefficient for each LCLU using a genetic algorithm to minimize the squared errors between observed and estimated runoff values using the rational method (Wang, 1991).

The reference evapotranspiration ($E_{\mathrm{To}}$) was calculated on a daily-basis using the Penman-Monteith equation parameterized by FAO 56 methodology (Allen et al., 1998). Afterwards, the evapotranspiration ($E_{\mathrm{T}}$) values were obtained for pasture and sugarcane land uses using Eq. 2.

$$E_{\mathrm{T}} = K_{\mathrm{S}} \cdot E_{\mathrm{Tc}} = K_{\mathrm{S}} \cdot (K_{\mathrm{C}} \cdot E_{\mathrm{To}}) \tag{2}$$

Where: $E_T$ is the real evapotranspiration (mm d$^{-1}$); Ks is the water stress coefficient (dimensionless); $E_{Tc}$ is the crop evapotranspiration (mm d$^{-1}$); Kc is the crop coefficient (dimensionless); and $E_{To}$ is the reference evapotranspiration (mm d$^{-1}$) (Eq. 3):

$$5 \quad E_{To} = \frac{0.408 \cdot \Delta \cdot (R_n - G) + \frac{\gamma \cdot 900}{(T_{avg} + 273)} \cdot U_2 \cdot (e_s - e_a)}{\Delta + \gamma \cdot (1 + 0.34 \cdot U_2)} \tag{3}$$

Where: $s$ is the slope vapor pressure (kPa °C$^{-1}$); $R_n$ is the net radiation (MJ m$^{-2}$ d$^{-1}$); $G$ is the soil heat flux (MJ m$^{-2}$ d$^{-1}$); $\gamma$ is the psychometric constant (kPa °C$^{-1}$); $U_2$ is the wind velocity at 2 m height (m s$^{-1}$); 900 is the approximate value of all the equations´ constants (kJ$^{-1}$ kg K d$^{-2}$); $T_{avg}$ is the average daily temperature (°C); $e_s$ is the saturated vapor pressure (kPa); and $e_a$
10    is the actual vapor pressure (kPa).

The water stress coefficient (Eq. 4) was calculated for each day i throughout the monitoring period. It uses the soil moisture monitored by the Frequency Domain Ratio (FDR) probes as the main input. When there is no or limited water available for the plants' transpiration, $K_S < 1$. Whenever the soil has water readily available for plant consumption, $K_S = 1$.

$$15 \quad K_{Si} = \frac{T_{AW} - D_{r_i}}{(1 - p) \cdot T_{AW}} \tag{4}$$

Where: $T_{AW}$ is the total available water in the soil root zone (mm) (Eq. 5); $D_{r_i}$ is the water depletion in the soil root zone (mm) (Eq. 6) on day $i$; and $p$ is the $T_{AW}$ fraction that the crop roots can extract from the soil without suffering from water stress (dimensionless).

$$20 \quad T_{AW} = 1000 \cdot (\theta_{fc} - \theta_{wp}) \cdot Z_f \tag{5}$$

Where: $\theta_{fc}$ is the soil field capacity (dimensionless); $\theta_{wp}$ is the soil wilting point; and $Z_f$ is the root zone depth.

$$D_{r_i} = 1000 \cdot (\theta_{fc} - \theta_i) \cdot Z_f \tag{6}$$

Where: $\theta_{fc}$ is the soil field capacity (dimensionless); $\theta_i$ is the average soil moisture along the root zone (measured by FDR probes); and $Z_f$ is the root zone depth.

All the necessary data and coefficients to calculate the water stress coefficient ($K_S$) and the adjusted evapotranspiration rate ($E_T$) are given in Table 2. It is worth mentioning that laboratory tests determined the soil field capacity and soil wilting
30    point using Büchner funnels and Richards extraction chambers (Richards, 1931). The soil samples were collected with undisturbed structure in volumetric rings at depths of 20, 50 and 100 cm.

**Insert Table 2**

The wooded Cerrado evapotranspiration ($E_T$) rates were estimated using the Priestley and Taylor (1972) method (Eq. 7). This method was chosen due to its simplicity to calculate the energy balance for the study area and the fact that is suitable for the instrumentation available. The Priestley and Taylor coefficients ($\alpha$) used to calculate the evapotranspiration were based on Cabral et al. (2015) measurements for a similar wooded Cerrado fragment located approximately 60 km away from the study site. The Priestley and Taylor coefficients ($\alpha$) differed according to the season: 1.09 for Summer (December – March); 1.00 for Fall (March – June); 0.77 for Winter (June – September); and 0.98 for Spring (September – December).

$$E_T = \alpha \cdot \left(\frac{1}{\lambda}\right) \cdot \left[\frac{s \cdot (R_n - G)}{s + \gamma}\right] \tag{7}$$

Where: $\alpha$ is the Priestly and Taylor coefficient (dimensionless), $\lambda$ is the latent heat of vaporization (MJ m$^{-2}$ d$^{-1}$); $s$ is the slope vapor pressure (kPa °C$^{-1}$); $R_n$ is the net radiation (MJ m$^{-2}$ d$^{-1}$); $G$ is the soil heat flux (MJ m$^{-2}$ d$^{-1}$); and $\gamma$ is the psychometric constant (kPa °C$^{-1}$).

The bare soil condition has no vegetation and, consequently, there is no transpiration. Thus, we applied the method developed by Ritchie (1972). This method has two phases: firstly, the soil evaporation is equal to the potential soil evaporation estimated using the Priestley and Taylor (1972) method adapted to free surfaces. During this phase, there is no water restriction (precipitation higher than evaporation) and the evaporation is governed by the available energy; secondly, the accumulated soil evaporation exceeds the precipitation and the soil evaporation is currently given as a function of the dry days that followed the last wet day. The evaporation cycle is interrupted and returns to the first phase whenever the precipitation exceeds the accumulated evaporation during the second phase.

## 2.4 Groundwater table fluctuation

The water table was registered twice a day (at 6 am and 6 pm) using pressure transducers (Diver, Schlumberger) placed inside two monitoring wells (well 1 located in the pasture area; and well 2 located inside the wooded Cerrado area). In the study site, both wells presented similar hydraulic conductivity according to the slug test (Bouwer and Rice, 1976) previously performed. We evaluated the aquifer hydraulic conductivity from both wells in order to validate the water table comparison among each other, as whether the aquifer condition in the wells were different, such comparison would not be fair. Both wells reach the water table at approximately 40 m depth in an unconfined sandstone formation (Botucatu formation), which belongs to São Bento Group of the Mesozoic age. Furthermore, the soils above the aquifer that appears thought the unsaturated zone are Cenozoic sediments weathered from the sandstone (Wendland et al., 2007). Additionally, despite the limited number of wells, the experimental design allowed a first look to the groundwater table behavior under different LCLU (pasture and wooded Cerrado) and a crosscheck with the surface water balance outcomes.

## 2.5 Data analysis

The normality assumption was tested using the Shapiro-Wilk test using a 95% confidence interval for rainfall, evapotranspiration, surface runoff and water balance residual datasets. The one-way analysis of variance (ANOVA) was applied to test the null and alternative hypothesis, that is, equality of surface runoff, evapotranspiration and water balance residual distribution functions between the four treatments (LCLU) versus the difference in distribution functions between at least two treatments. Additionally, the multiple comparisons between treatments were performed using the Tukey test (Montgomery, 2008). The rainfall, evapotranspiration, surface runoff, water balance residual and soil moisture graphs were plotted using a daily basis timescale. The groundwater table fluctuation was plotted using a monthly timescale due to the noise typically found in this kind of measurement. In order to present the order of magnitude along the years, the data was also resumed annually in tables and figures.

## 2.6 Data uncertainties

Data uncertainties are flaws found in the available information used to represent the reality and basically depend on the knowledge of the observed data (Refsgaard et al., 2007). The water balance uncertainty can be given by the standard error propagated from its components. Thus, the water balance residual ($dS/dt$) standard error was propagated (Eq. 8).

$$\sigma_{ds/dt} = \left(\sigma_P^2 + \sigma_{O_F}^2 + \sigma_{E_T}^2\right)^{0.5} \tag{8}$$

Where: $\sigma$ is the standard error (mm yr$^{-1}$); $dS/dt$ is the water balance residual; $P$ is rainfall; $O_F$ is surface runoff; $E_T$ is evapotranspiration.

The absolute error of the rainfall estimations was calculated using Eq. 9 based on the instrument accuracy informed by the manufacturer. Considering the used tipping bucket rain gauge (TB4), the error may reach up to ±3%.

$$\sigma_P = \bar{P} \cdot \varepsilon_P \tag{9}$$

Where: $\sigma_P$ is the standard error for rainfall (mm yr$^{-1}$); $\bar{P}$ is the annual average rainfall (mm yr$^{-1}$); $\varepsilon_P$ is the relative error from the tipping bucket rain gauge informed by the manufacturer.

The surface runoff may vary due to the heterogeneities found between plots' replicates (Wendt et al., 1986; Nearing et al., 1999; Gómez et al., 2001; Sadeghi et al., 2013). Thus, the standard error for surface runoff is given by the standard deviation of the replicated plots (Eq. 10).

$$\sigma_{O_F} = \left[\frac{1}{N-1}\sum_{i=1}^{N}\left(O_{F_i} - \overline{O_F}\right)^2\right]^{0.5} \tag{10}$$

Where: $\sigma_{O_F}$ is the standard error for surface runoff (mm yr$^{-1}$); N is the number of observations; $O_{F_i}$ is the surface runoff observed in plot $i$ (mm yr$^{-1}$); $\overline{O_F}$ is the average surface runoff between plots (mm yr$^{-1}$).

As previously mentioned, the evapotranspiration was estimated using the FAO 56 methodology (Allen et al., 1998) for pasture and sugarcane, and Priestley and Taylor (1972) for wooded Cerrado and bare soil. However, this study does not have evapotranspiration observations to evaluate how well this variable was estimated in the study sites. Consequently, the uncertainties for evapotranspiration estimates were calculated by combining all input variable uncertainties (Eq. 11) which were measured in the field for the FAO 56 method (temperature, relative humidity, solar radiation, barometric pressure and soil moisture) and the Priestley and Taylor method (temperature, net radiation and soil heat flux).

$$\sigma_{E_T} = \left[ \sum_{i=1}^{N} \left( \frac{\partial \text{var}_i}{\partial_{E_T}} \cdot u_{\text{var}_i} \right)^2 \right]^{0.5} \tag{11}$$

Where: $\sigma_{E_T}$ is the standard error for evapotranspiration (mm d$^{-1}$); var$_i$ is the measured input variable $i$; $u$ is the uncertainty of variable $i$.

## 3 Results and discussion

### 3.1 Water balance

The water balance components show different patterns according to the LCLU (Table 3 and Fig. 3). We verified this using a multiple comparison test (Tukey) in which the water balance residual variation (d$S$/d$t$) in the wooded Cerrado was statistically different from the other LCLU (pasture, sugarcane and bare soil), which presented similar means. The annual evapotranspiration in the wooded Cerrado was the highest among the analyzed LCLU. The pasture presented similar annual evapotranspiration values to those found in the sugarcane and bare soil. However, sugarcane and bare soil had different means for evapotranspiration among them. Concerning the surface runoff, bare soil and pasture presented significant differences from the other LCLU. Hence, the results agree with previous studies, where the land use presented regulatory functions in the water balance (Krishnaswamy et al., 2013; Nobrega et al., 2017). Daily data of the water balance components and average soil moisture (first meter of soil) from the analyzed LCLU are available as supplement (S1).

### Insert Table 3

### Insert Fig. 3

The average annual rainfall in the study site was 1388 mm yr$^{-1}$ between 2012 and 2016 (Fig. 3A). It was approximately 100 mm yr$^{-1}$ lower than the average observed during the last 37 years (Cabrera et al., 2016) due to a drought in 2014 (Getirana, 2015; Melo et al., 2016). Additionally, precipitation (P) and water balance residual (d$S$/d$t$) were the water balance

components that had the largest variations throughout the monitoring period. We observed that sugarcane and pasture were similar considering the water balance components' patterns and orders of magnitude. However, wooded Cerrado and bare soil presented different characteristics from the other LCLU.

The evapotranspiration estimates (Fig. 3B) were different between the considered land covers (Table 3). We observed that the wooded Cerrado evapotranspiration presented the highest rates among all analyzed LCLU and also the smallest variability throughout the year. However, there is no agreement between the measurements and estimates performed in wooded Cerrado areas and the present study (Table 4), due to the diverse rainfall patterns among the study sites and the different methods used to measure or estimate the evapotranspiration. Thus, this study shows evidence of the need for reference values of evapotranspiration in undisturbed areas for a better understanding of their role in the water cycle. In addition, the rainfall is different for the studies compared in Table 4.

**Insert Table 4**

The sugarcane evapotranspiration rates were higher than pasture due to the higher crop water demand during the initial phase of its annual cycle, and both evapotranspiration estimates agree with measurements performed by previous studies (Sakai et al., 2004; Cabral et al., 2012; Nobrega et al., 2017). Both sugarcane and pastureland ceased the evapotranspiration during the dry season due to the decoupling condition, when the radiation is the only contributor to the evapotranspiration process (water stress condition) (Pereira, 2004). This happens due to the lack of water readily available for the plants along the root zone. This condition does not repeat in the wooded Cerrado as its root system is deeper (Canadell et al., 1996), and the plants may reach water in depths where sugarcane and pasture cannot do this. However, the root zone depth of an undisturbed vegetation such as the wooded Cerrado is uncertain and may vary according to the soil characteristics (Canadell et al., 1996) and the groundwater table level (Leite et al., 2018). It may influence the plants' transpiration (Rawitscher, 1948; Oliveira et al., 2005), and consequently the water balance residual.

The surface runoff (Fig. 3C) presented the lowest values among all water balance components and it was the most influenced variable by the land use. There was no surface runoff generation during August due to the non-occurrence of rainfall events. The bare soil was the most sensitive to the runoff generation throughout the dry season, when the occurrence of rainfall events was lower. The months of January, February and March registered the highest averages for rainfall due to the wet season. The runoff coefficients for wooded Cerrado, sugarcane, pasture, and bare soil were 0.001, 0.007, 0.029 and 0.063, respectively. These values are in compliance with a previous study performed by Oliveira et al. (2016) for similar conditions. Although a higher runoff coefficient for sugarcane in comparison with the pasture was expected, sugarcane presented runoff values significantly lower than pasture as the soil tillage increased water infiltration and, consequently, reduced the surface runoff. Additionally, there is an apparent inverse linear relationship between culture size and runoff coefficient.

The surface runoff in sugarcane plantations is not well understood by the scientific community despite the economic importance to the country. The monitored values agreed with previous studies in the same study area (Oliveira et al., 2016;

Youlton et al., 2016b; Anache et al., 2018). The highest runoff rates were registered after planting and harvesting events, when the soil is more exposed to the rainfall.

In addition, the surface runoff in the pasture presented significantly higher rates in comparison with sugarcane and wooded Cerrado, due to the top soil compaction caused by grazing. The order of magnitude of the measured runoff in the pasture plots are similar to those previously published (Saraiva et al., 1981; Silva et al., 2011; Dedecek, 1989). The high runoff variability observed in this land use was due to the variable presence of animals, which were managed in an extensive approach and, consequently, heterogeneities may happen in soil compaction and vegetation conditions (Nacinovic et al., 2014). Additionally, the deforestation and agricultural land uses may increase soil compaction, as the LCLUC influence the hydrological patterns along the soil profile by evident modifications in the soil characteristics (bulk density, infiltration capacity, etc.) (Lamparter et al., 2016; Meister et al., 2017; de Almeida et al., 2018).

The wooded Cerrado presented the lowest runoff rates among all LCLU due to the higher soil protection, which avoided the overland flow generation. The order of magnitude of the surface runoff in the wooded Cerrado was similar to those found in shrublands and forested areas (Dedecek, 1989; Silva et al., 2011; Oliveira, 2012; Nacinovic et al., 2014; Oliveira  et al., 2015). These reduced surface runoff rates in the wooded Cerrado increase soil water infiltration in comparison with pasture, sugarcane and bare soil. Thus, higher infiltration rates increase plant water availability (Krishnaswamy et al., 2013). Additionally, the organic matter layer above the soil (wooded Cerrado's forest floor litter) reduces the overland flow. In some cases, the litter removal may increase the surface runoff up to 50% (Gomyo and Kuraji, 2016).

The water balance residuals (Fig. 3D) showed the water surplus (positive values) and deficits (negative values), evidencing the consumption of the soil water storage along the dry season (June – September). Such behavior is also evidenced by the average soil water content along the first meter of soil (Fig. 3E), as the soil moisture became lower in the wooded Cerrado in comparison with pasture and sugarcane observations during the dry season. This is because only the wooded Cerrado condition had negative values during the dry season, as the other land uses (sugarcane and pasture) had no conditions to remove water from deeper regions due to the shallow root system and their physiological characteristics. Thus, wooded Cerrado vegetation adapted to dry weather conditions, due to the plant water demand for evapotranspiration, which reduced gradually as long as the water balance residual accumulated during the wet season was consumed (Oishi et al., 2010; Christoffersen et al., 2014; Cabral et al., 2015; Oliveira  et al., 2015). In addition, the structural quality of soil from undisturbed woodlands leads to a higher capacity to retain water than that pasture soil (Tseng et al., 2018).

## 3.2 Groundwater table fluctuation

The groundwater table fluctuation inside the wooded Cerrado area was lower compared to the pasture area considering the observations of the monitoring wells (Fig. 3F). This variation in the wooded Cerrado was less than 1 m per year due to the water deficit periods during the dry season. In addition, a similar study in a Cerrado area (Villalobos-Vega et al., 2014) verified that the groundwater table fluctuation tend to be lower where the unsaturated zone is thicker. In the well located in the pasture, the water table fluctuated negatively along 2014 and 2015 due to the drought that happened in 2014 (Getirana,

2015). The water surplus of 2015-2016 happened due to the La Niña phenomena, that raised the rainfall pattern after the long dry season of 2014-2015 (Kakatkar et al., 2018). Consequently, the water table raised along 2016. The water scarcity periods that occurred in the wooded Cerrado area were due to the higher vegetation demand, as it is denser than the pasture and it has a deep root system. It is important to remember that both monitoring wells (pasture and wooded Cerrado) have the same non-saturated zone thickness (40 m) and similar hydraulic characteristics. However, in order to perform more complete evaluations of the hydrogeological processes in the study site, further measurements and additional monitoring wells may be necessary.

There are clear evidences that a time lapse between the water infiltration thought the soil and the aquifer recharge exists due to the huge non-saturated zone thickness (around 40 m) (Fig. 3F). For this reason, we cannot ignore that the evapotranspiration influences the aquifer recharge, as previously reported by other studies under similar and different conditions from the present study (Scanlon et al., 2005; Scott et al., 2014; Lucas et al., 2015; Oliveira et al., 2017; Lucas and Wendland, 2015). Changes in the LCLU, such as the potential conversion from wooded Cerrado to an agricultural LCLU (here we tested the pasture), may affect groundwater recharge, processes and availability. These effects tend to be non-linear and difficult to analyze as they result from complex interactions between LCLU and hydrological processes (Han et al., 2017). Thus, mathematical approaches (Archer and Fowler, 2018; Gómez et al., 2018) or natural tracers (Su et al., 2018) are useful tools to verify the response time of the groundwater table to the water balance from different LCLU. In further studies, such techniques may be part of a solution to investigate how responsive the aquifer is to the surface water partitioning in the conditions considered here along the time.

The evapotranspiration and root zone depth controlled the water balance residual and, consequently, the water percolation throughout the non-saturated zone and aquifer recharge (Finch, 1998; Gouvêa and Wendland, 2011; Krishnaswamy et al., 2013; Lucas and Wendland, 2015; Domínguez et al., 2016; Manzione et al., 2017). In the pasture area (site 1), the soil water that was not consumed by the plants and neither evaporated, flowing down along the unsaturated zone. Consequently, the water uptake by the plants becomes unfeasible as the roots are more shallow (see Table 2) than in the wooded Cerrado. Furthermore, the aquifer recharge decreases as the vegetation density increases in undisturbed Cerrado areas (Oliveira et al., 2017), following the principle that the increased canopy cover of the wooded Cerrado found in the study area may occur due to the deep groundwater level (Leite et al., 2018; Villalobos-Vega et al., 2014). However, the water balance analysis performed here focused on hillslope hydrology, and the monitoring wells depths reflects the aquifer behavior in a broader area covered by pasture and wooded Cerrado in comparison with the 100-m2 plots where we monitored the surface water partitioning. Thus, all assumptions made throughout this section are subject to further analysis, which may include the water balance calculation for a broader area (e. g. the whole 300 ha wooded Cerrado fragment where part of the present study plots were located to represent either LCLU). Nevertheless, the hillslope scale water balance outcomes are comparable to the groundwater table fluctuations, as the wells represent the plots' surroundings for the pasture and wooded Cerrado LCLU. Groundwater depth monthly datasets of the monitoring wells (wooded Cerrado and pasture) are available as supplement (S2).

### 3.3 Data uncertainties

The water balance components ($P$, $O_F$, $E_T$, and d$S$/d$t$) uncertainties were calculated for each land cover (Table 5) and the relative errors agreed with previous water balance studies in different scales (Graham et al., 2010; Oliveira et al., 2014). Concerning rainfall (P), which is the only water input, the same measured values were used for all LCLU. The surface runoff presented higher relative uncertainties in the pasture plots, followed by bare soil, wooded Cerrado, and later by sugarcane. Evapotranspiration (ET) can be a potential source of uncertainties in a water balance (da Paz et al., 2014), it was estimated adopting methods that use ground measured variables (e.g. temperature, relative humidity, solar radiation and soil moisture) and the relative uncertainties may reach up to 63%. The pasture presented the higher relative error for ET due to the reduced average evapotranspiration compared to the other LCLU (sugarcane and wooded Cerrado), and the wooded Cerrado presented the higher standard error for ET. Nevertheless, the water balance hypothesis mainly relies on minimizing uncertainties in the evapotranspiration estimates or measurements (Beven, 2006). The surface runoff ($O_F$) did not contribute significantly in the water balance error propagation due to its reduced order of magnitude compared to the other water balance components ($E_T$ and $P$). However, runoff measurements produced relative errors that reached up to 26% (pasture).

**Insert Table 5**

**Insert Fig. 4**

The water balance residual (d$S$/d$t$) presented the accumulated uncertainties from rainfall, surface runoff and evapotranspiration (Fig. 4). All LCLU presented standard errors for d$S$/d$t$ with similar orders of magnitudes, except the bare soil, which presented lower errors due to the reduced number of inputs to calculate the evaporation in an open-surface condition. The wooded Cerrado accumulated an error of 636 mm yr$^{-1}$ for the water balance residual mainly due to its highest component: evapotranspiration. This suggests that efforts to minimize the uncertainties in measuring or estimating the evapotranspiration in the wooded Cerrado may significantly improve its water balance. Additionally, other land uses (pasture and sugarcane) also presented a high d$S$/d$t$ uncertainty due to the standard error accumulated from the evapotranspiration estimates.

### 3.4 Water balance trade-offs due to the LCLUC

The Brazilian Cerrado is very important economically, as it is responsible for most of the agricultural production that supplies both external and internal markets (Klink and Machado, 2005). A better understanding of the trade-offs between the ecosystem and economical needs that govern the land cover and land use dynamics in the Cerrado biome is necessary (Marris, 2005). The undisturbed Cerrado area was reduced to 50% of its original extension due to intense land use (Lapola et al., 2013; Alkimim et al., 2015). The undisturbed vegetation helps to maintain the water cycle, with low surface runoff and high evapotranspiration rates.

This study show evidence that the suppression of the wooded Cerrado and the conversion to agricultural land uses, such as sugarcane and pasture, increased the surface runoff and decreased evapotranspiration, even considering measurements and

estimation method uncertainties. Consequently, the water balance residual (d$S$/d$t$), increased significantly, suggesting that the infiltration also rose. However, the soil water was not available to the plants' roots in the agricultural land uses during the dry season (April – September), likewise the wooded Cerrado, where the soil water content is (in average) higher than the agricultural LCLU (Fig. 3E). Thus, the percolation and aquifer recharge increased in the agricultural area (site 1) and this fact was explained by the water table fluctuations observed in the monitoring well located in the pasture that were not observed in the monitoring well located inside the wooded Cerrado area (site 2).

Previous studies show that native forests help to maintain aquifer recharge by the high infiltration rates, similarly to observations performed in tropical forests in south India and in mountainous areas in the Himalayas (Krishnaswamy et al., 2013; Ghimire et al., 2014). Nevertheless, evapotranspiration appears as a key component in the aquifer recharge control in the study site condition, which is located in a Guarani Aquifer System outcrop zone (Lucas et al., 2015; Lucas and Wendland, 2015). Therefore, the aquifer recharge rates, evidenced here by the groundwater table fluctuation (Fig. 3F), may be reduced in forested areas in comparison with agricultural landscapes due to the atmospheric and vegetation water demands, and the increased soil water retention capacity (Adane et al., 2018; Dias et al., 2015; Wang et al., 2018; Tseng et al., 2018). This validate the information that the LCLU significantly impacts groundwater recharge (Scanlon et al., 2005; Scott et al., 2014; Lucas et al., 2015; Dawes et al., 2012). This fact may suggest that the ecosystem services of native forests, such as the wooded Cerrado, is not the aquifer recharge maintenance, but rather the constant return of water to the atmosphere throughout the year.

Significant and non-significant changes in the water partitioning may be observed in the case of substituting the wooded Cerrado by bare soil, pasture or sugarcane (Fig. 5). We highlight the $E_T$ reduction (more than 400 mm yr$^{-1}$). The worst case scenario (bare soil condition) is generally the transition between the LCLU (fallow condition), increasing significantly surface runoff rates and impacting soil and water conservation (Pimentel et al., 1995).

**Insert Fig. 5**

Sugarcane and pasture presented trade-offs that were equal, as non-significant changes occurred among each other's water balance components. However, the sugarcane plantation presented a higher potential to maintain the evapotranspiration rates closer to the undisturbed Cerrado conditions than the pasture, agreeing with previous estimates (Loarie et al., 2011). In addition, it is suggested that sugarcane has increased evapotranspiration rates in comparison with other annual crops (Guarenghi and Walter, 2016; Hernandes et al., 2018b, a).

**4 Conclusions**

This paper presented an experimental approach at the hillslope scale concerning the possible water partitioning trade-offs due to the LCLUC dynamics. We monitored the water balance components over 5-years in different land covers: wooded Cerrado, pasture, sugarcane and bare soil. These land covers are subjected to the current LCLUC dynamics in southeastern Brazil. The water partitioning observations in different LCLU confirm that modifications in the land surface

conditions may significantly change the water balance residual (up to 584 mm yr$^{-1}$). The decrease in evapotranspiration and increase in surface runoff are common patterns when the wooded Cerrado is replaced by agricultural land uses. The water balance outcomes evidences that the undisturbed Cerrado vegetation consumes the soil water storage along the dry season (June – September). By contrast, the agricultural LCLU (pasture and sugarcane) reduce or even stop the water consumption

along either season. The probable main reason for that is the reduced water retention capacity commonly found in disturbed soils.

Higher water consumption by dense and native vegetation, such as the wooded Cerrado, also happens due to the higher infiltration rates, increasing the plant water availability. In general, the root systems are deeper than pasturelands and sugarcane plantations, and have the capacity of reaching water for transpiration deeper in the soil profile. This maintains

evapotranspiration throughout the year, even during the dry season. Hence, less water becomes available for the aquifer recharge in areas where the canopy layer is higher and denser. However, reference values for evapotranspiration in undisturbed land covers, such as the wooded Cerrado, are still needed in order to reduce uncertainties from the current approximations and validate the water balance hypothesis. Particularly the undisturbed Cerrado and agricultural LCLU should be investigated concerning its potential to maintain the aquifer recharge and groundwater availability, as well as

answering how responsive the aquifer is to the surface water partitioning in different LCLU along the time.

**Data availability**

The datasets underlying this research are available as supplements (S1 and S2) of this paper and they are accessible from the following data repository link: http://www.hydroshare.org/resource/a1c032dbb78d48748b673c876c20b21c.

**Author contribution**

JAAA, EW, PTSO and CY designed the experiments and JAAA, LMPR, PTSO and CY carried them out. JAAA, LMPR and EW analysed the experiments' outcomes and discussed the results. JAAA and EW prepared the manuscript with contributions from all co-authors.

**Competing interests**

The authors declare that they have no conflict of interest.

**Acknowledgements**

This study was supported by grants from the Ministry of Science, Technology, Innovation and Communication – MCTIC and the National Council for Scientific and Technological Development – CNPq (grant numbers 201109/2015-8,

142393/2015-0, 150057/2018-0, 441289/2017-7 and 306830/2017-5), the São Paulo Research Foundation — FAPESP (grant number 2015/03806-1), and the Coordination of Improvement of Higher Education Personnel – CAPES (finance code 001). The authors acknowledge the Graduate Program in Hydraulics and Sanitary Engineering – PPGSHS (USP-EESC) for the scientific support, and the Arruda Botelho Institute – IAB for allowing the development of this study inside its private land. The authors would like also to thank the editor and the anonymous referees for their useful comments, which substantially improved the manuscript.

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

**Tables**

Table 1: Monitoring site instrumentation characteristics.

| Variable | Sensor | Height or depth (m) | Measurement range | Maximum error | Site no. |
|---|---|---|---|---|---|
| Temperature (ºC) and relative humidity (%) | HMP45C | 2.0 | -39.2 to +60°C and 0.8% to 100% | ±0.5°C and ±3% | 1 |
| | HC2S3 | 9.5 and 11.0 | | ±0.1°C and ±0.8% | 2 |
| Rainfall (mm) | Hydrological Services TB4 | 1.5 and 11.0 | 0 to 700 mm h$^{-1}$ | ±3% | 1 and 2 |
| Atmospheric pressure (mbar) | Vaisala CS106 | 1.0 | 500 to 1100 mBar | ±1.5 mBar | 1 |
| Wind direction (°) and velocity (m.s$^{-1}$) | Young 05103 | 2.0 and 11.0 | 0 to 360° and 0 to 100 m s$^{-1}$ | ±3° and ±0.3 m s$^{-1}$ | 1 and 2 |
| Solar radiation (MJ.m$^{-2}$) | Kipp & Zonen CMP3 | 2.0 | 0 to 2000 W m$^{-2}$ | ±5% | 1 |
| Soil moisture (%) | FDR | -0.3; -0.6 and -0.9 | 0 to ~65% | ±3% | 1* |
| | Environscan Sentek | -0.1; -0.5; -0.7; -1.0 and -1.5 | | | 2 |
| Net solar radiation (W.m$^{-2}$) | Kipp & Zonen NR-LITE2 | 11.0 | ±2000 W m$^{-2}$ | ±5% | 2 |
| Soil heat flux (W.m$^{-2}$) | Hukseflux HFP01 | -0.1 | ±2000 W m$^{-2}$ | -15% to +5% | 2 |
| Groundwater table (m) | Diver Schlumberger | -40.0 and -39.2 | 0 to 10 m | ± 2.5 cm | 1 and 2 |

*At site 1, the bare soil did not have soil moisture probes.

**Table 2: Variables used to calculate evapotranspiration for sugarcane and pasture.**

| Variable | Sugarcane | Pasture | Source |
|---|---|---|---|
| $Z_f$ (m) | 1.2 – 2.0 | 0.5 – 1.5 | Allen et al. (1998) |
| $p$ | 0.65 | 0.60 | |
| $\theta_{fc}$ | 0.14 | 0.14 | Values obtained from laboratory essays |
| $\theta_{wp}$ | 0.09 | 0.09 | (Oliveira, 2014) |
| $K_C$ | Plant (1): 0.50[a]; 0.80[b]; 095[c]; 1.10[d]; 1.18[e]; 0.92[f]; 0.68[g] Ratoon (2): 0.55[a]; 0.80[b]; 090[c]; 1.00[d]; 1.05[e]; 0.80[f]; 0.60[g] | 0.75 (3) | (1) Doorenbos et al. (1975) (2) Doorenbos et al. (1979) (3) Allen et al. (1998) |

Approximate sugarcane age (days): a (0-30); b (30-60); c (60-75); d (75-120); e (120-300); f (300-330), g (330-360).

**Table 3: Mean and standard deviation of the annual water balance components for 2012-2016 period.**

| Land use | Water balance components (mm yr$^{-1}$) | | | |
|---|---|---|---|---|
| | $P$ | d$S$/d$t$ | $E_T$ | $O_F$ |
| Wooded Cerrado | | 185±182b | 1201±49a | 2±2c |
| Pasture | 1388±188 | 689±135a | 654±137bc | 45±26b |
| Sugarcane | | 571±100a | 801±212b | 16±18c |
| Bare soil | | 769±96a | 482±40c | 137±62a |
| Data source | observations | residuals | estimations | observations |

d$S$/d$t$: water balance residual; $P$: precipitation; $E_T$: evapotranspiration; $O_F$: surface runoff; identical lower case letters indicate no significant difference between means in the same column by the Tukey means comparison test (P value > 0.05).

**Table 4: Evapotranspiration and rainfall from studies developed in Brazil under similar LCLU of the present study.**

| Land use | Lat | Long | Method | $E_T$ (mm yr$^{-1}$) | $P$ (mm yr$^{-1}$) | References |
|---|---|---|---|---|---|---|
| Sugarcane[1] | -21.63 | -47.78 | EC | 892 | 1194 | Cabral et al. (2012) |
| Sugarcane[2] | -21.63 | -47.78 | EC | 685 | 1353 | Cabral et al. (2012) |
| Sugarcane[1] | -22.18 | -47.85 | FAO 56 | 930 | 1535 | Present study |
| Sugarcane[2] | -22.18 | -47.85 | FAO 56 | 715 | 1290 | Present study |
| Wooded Cerrado | -21.62 | -47.65 | EC | 811 | 1498 | da Rocha et al. (2009) |
| Wooded Cerrado | -21.62 | -47.65 | EC | 1228 | 1448 | Cabral et al. (2015) |
| Wooded Cerrado | -15.93 | -47.88 | EC | 821 | 1440 | Giambelluca et al. (2009) |
| Wooded Cerrado | -15.80 | -55.33 | PM RS | 1004 | 1696 | Nobrega et al. (2017) |
| Wooded Cerrado | -22.18 | -47.85 | PM RS | 823 | 1194 | Oliveira et al. (2015) |
| Wooded Cerrado | -22.18 | -47.85 | PT | 1201 | 1388 | Present study |
| Pasture | -3.01 | -54.53 | EC | 647 | 1597 | Sakai et al. (2004) |
| Pasture | -15.81 | -55.34 | PM RS | 639 | 1780 | Nobrega et al. (2017) |
| Pasture | -22.18 | -47.85 | FAO 56 | 654 | 1388 | Present study |
| Cerrado and Amazon transition | -11.41 | -55.33 | EC | 1005 | 2000 | Vourlitis et al. (2002) |

Lat: Latitude; Long: Longitude; $E_T$: Evapotranspiration; $P$: Rainfall; EC: Eddy covariance; PM RS: Penman-Monteith and remote sensing; FAO 56: Allen et al. (1998); PT: Priestley and Taylor (1972); 1: Plant; 2: Ratoon.

**Table 5: Water balance uncertainties.**

| Variables | | LCLU | | | |
|---|---|---|---|---|---|
| | | Wooded Cerrado | Pasture | Sugarcane | Bare soil |
| $P$ (observed) | Average (mm yr$^{-1}$) | | | 1388 | |
| | Standard error ($\sigma$) (mm yr$^{-1}$) | | | 42 | |
| | Relative error ($\varepsilon$) (%) | | | 3% | |
| $O_F$ (observed) | Average (mm yr$^{-1}$) | 2 | 45 | 16 | 137 |
| | Standard error ($\sigma$) (mm yr$^{-1}$) | 0.2 | 12 | 1 | 27 |
| | Relative error ($\varepsilon$) (%) | 10% | 26% | 5% | 19% |
| $E_T$ (estimated) | Average (mm yr$^{-1}$) | 1201 | 654 | 801 | 482 |
| | Standard error ($\sigma$) (mm yr$^{-1}$) | 634 | 412 | 361 | 39 |
| | Relative error ($\varepsilon$) (%) | 53% | 63% | 45% | 8% |
| d$S$/d$t$ (residual) | Average (mm yr$^{-1}$) | 185 | 689 | 571 | 769 |
| | Standard error ($\sigma$) (mm yr$^{-1}$) | 636 | 415 | 363 | 63 |
| | Relative error ($\varepsilon$) (%) | 344% | 60% | 64% | 8% |

d$S$/d$t$: water balance residual; $P$: precipitation; $E_T$: evapotranspiration; $O_F$: surface runoff.

**Figures**

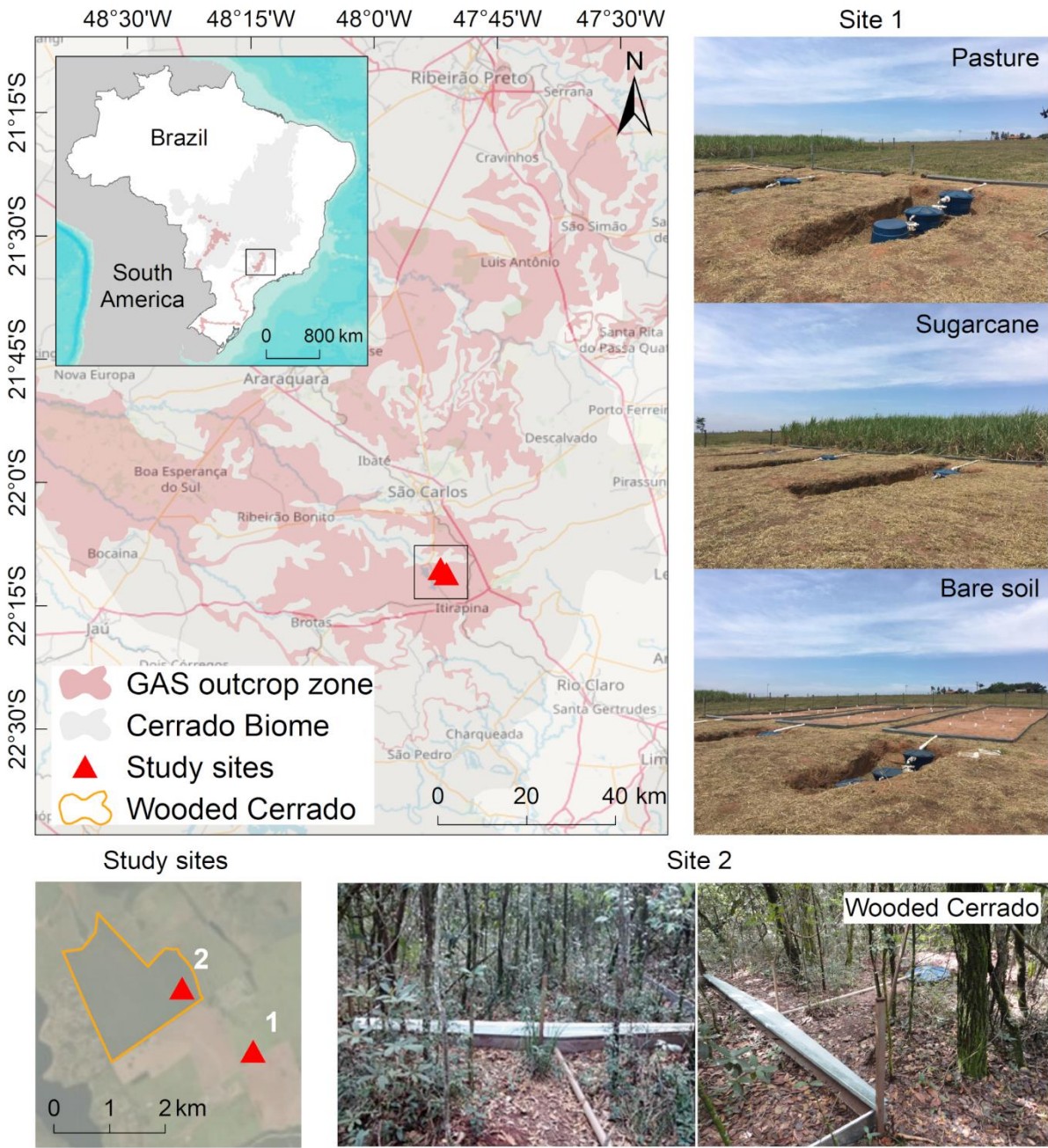

Figure 1: Location of study sites, Cerrado biome borders, Guarani Aquifer System (GAS) outcrop zone distribution in Brazil, and experimental design, where site 1 contains the plots with agricultural land uses (pasture, sugarcane and bare soil) and site 2 contains the plots with wooded Cerrado.

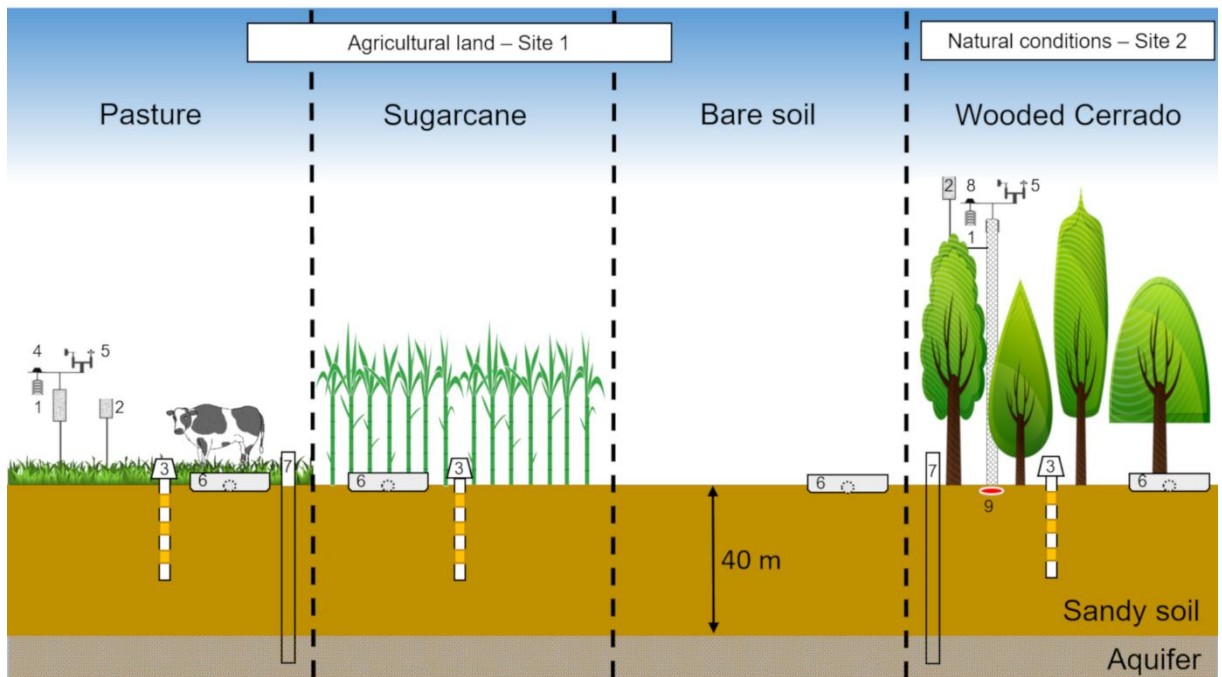

**Figure 2: Hydrological monitoring performed on the four treatments: (1) relative humidity and air temperature probes at 2 m (site 1) and 11 m (site 2); (2) rainfall gauges; (3) soil moisture sensors; (4) solar radiation sensor; (5) wind speed and direction (anemometers) at 2 m (site 1) and at 11 m (site 2); (6) surface runoff collectors; (7) monitoring wells equipped with water table pressure transducers; (8) net radiation sensor; (9) soil heat flux plate.**

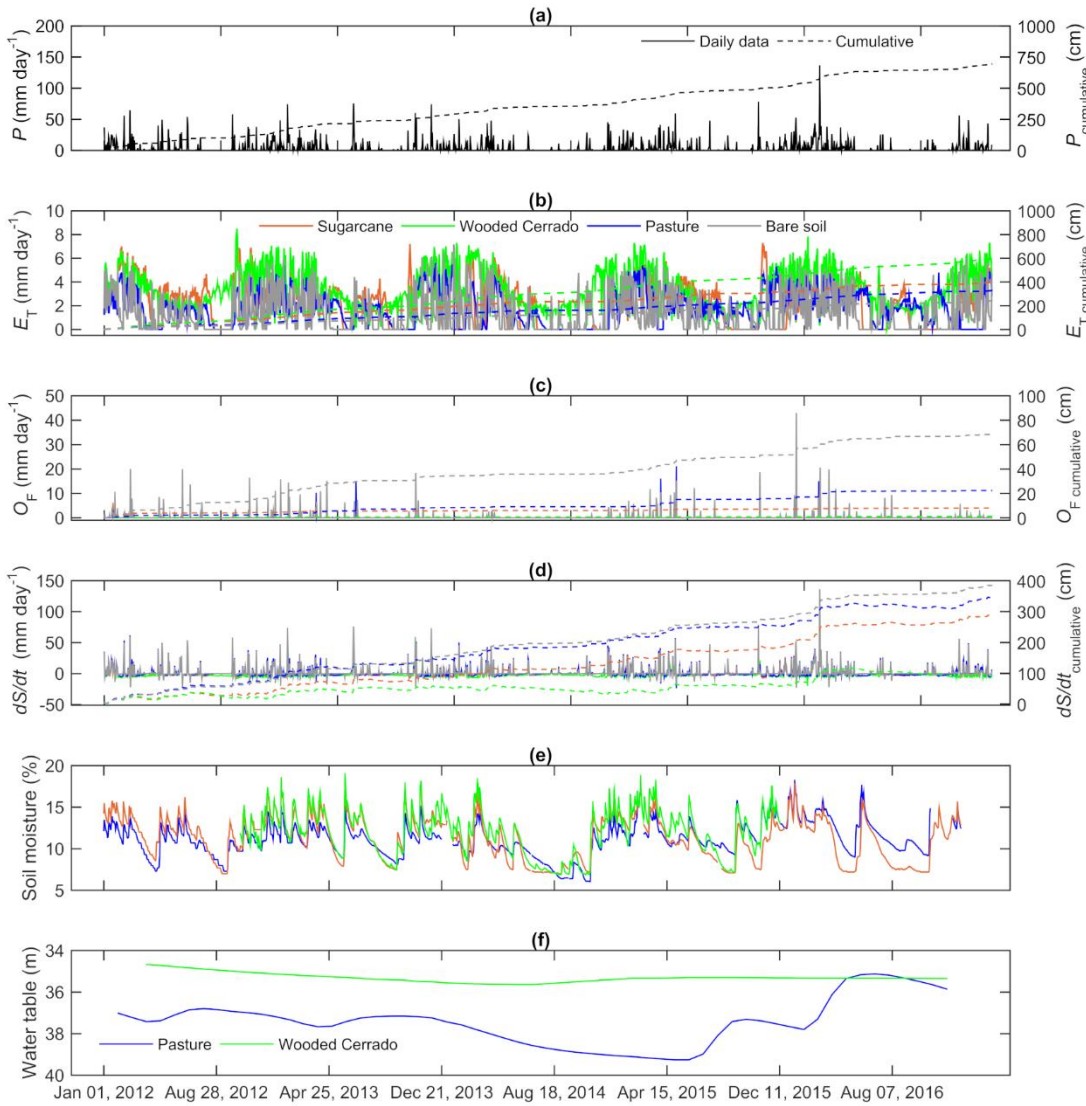

**Figure 3: Water balance components for different LCLU rainfall, *P* (a); evapotranspiration, *E*ᴛ (b); surface runoff, *O*ꜰ (c); water balance residual, d*S*/d*t* (d); Right axes present the cumulative sum of the variables represented by graphs (a), (b), (c) and (d). Soil moisture in the first meter of soil (e) for pasture, sugarcane and wooded Cerrado; and water table (f) depth of the monitoring wells located in site 1 (pasture) and site 2 (wooded Cerrado).**

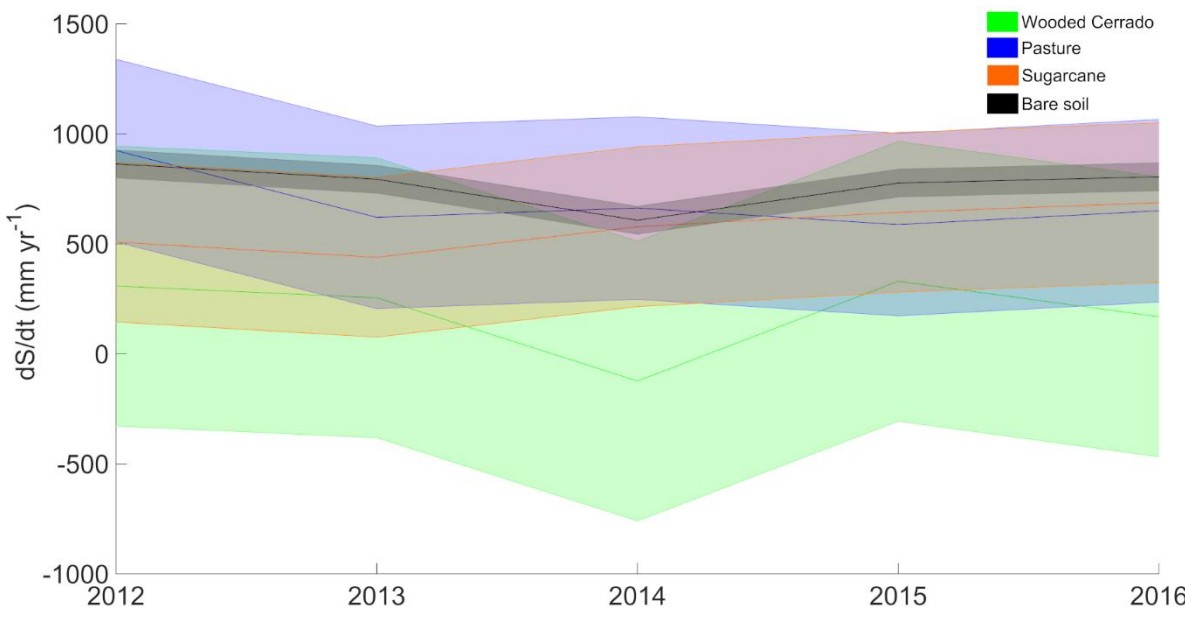

**Figure 4: Annual water balance residual (d*S/*d*t*) for different LCLU during 2012-2016 period; shaded areas indicate the standard error (uncertainties) of d*S/*d*t* estimates.**

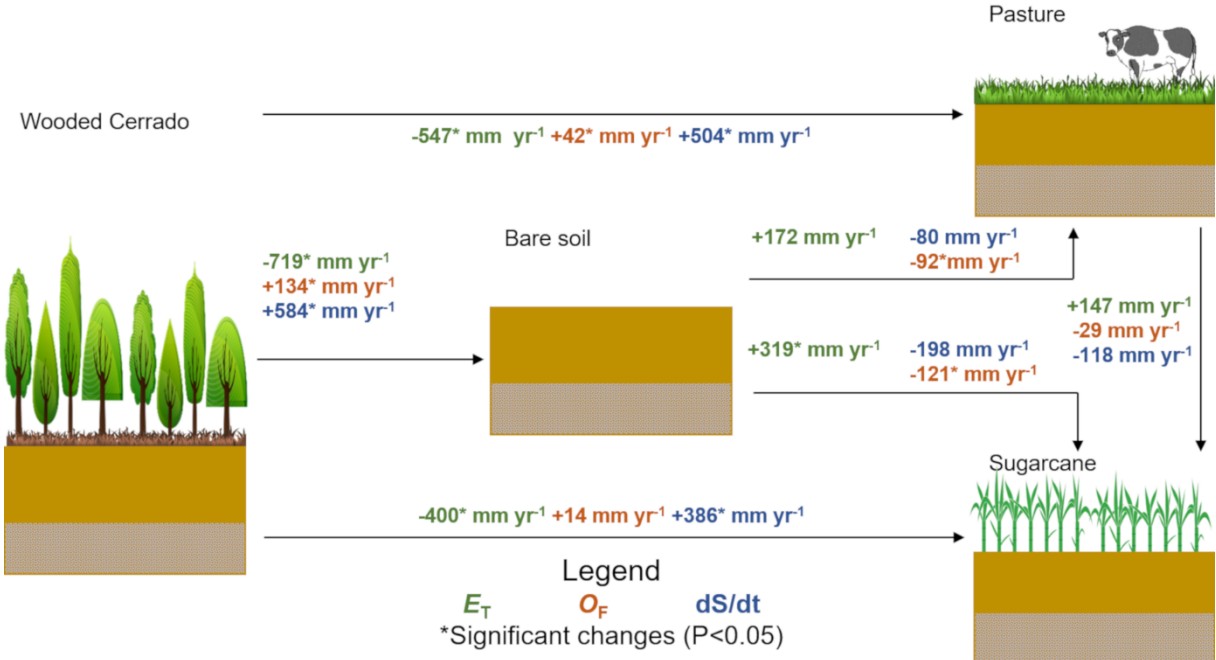

**Figure 5: Annual observed means (2012-2016) of hydrological trade-offs related to evapotranspiration ($E_T$), surface runoff ($O_F$), water balance residual (d$S$/d$t$) due to potential LCLUC found in southeastern Brazil.**