# Peer review of "Hydrological trade-offs due to different land covers and land uses in the Brazilian Cerrado"

_Hydrology and Earth System Sciences, 2018_

## Referee Comment (RC1) · Anonymous Referee #1 · 11 Sep 2018

**Review on HESS Discussion paper**
**„Hydrological trade-offs due to different land covers and land uses in the Brazilian Cerrado**

**By Anache, Wendland et al.**
* * *
Comments for the authors (anonym):

The paper treat an important topic in the frame of LCLU for the Cerrado of Brazil. Until now only very few studies with experimental site data (see Oliveira, Nobrega) cover the Cerrado Biome in Brazil (most deals with Amazon rainforest). Problem statement is clear and well written. In the space of row 3 – 10 an outlook on the process of further Cerrado conversion should be added and why sugarcane in the study area will be important in this process of LUC. The aim of the study is well written (row 25-28).
Experimental instrumentation is detailed described and adequate for the aim of longterm monitoring between the different land uses and Cerrado sensu stricto. In 2.2 following should be added for understanding the calculations later:
Page 3: Time interval of soil water measurements (daily?)
As basic information were $k_{sat}$ measurements done to understand the importance of infiltration to the groundwater of the Entisols?
Page 4 row 3: it seems better to define surface runoff as $Q_{sur}$ or $O_f$ (overland flow) instead of Q, because in most hydrological studies Q is defined as total discharge (see hydrological terms).
Evapotranspiration was calculated in the standard form on the base of Penman-Monteith (ETo). Water stress coefficient was calculated on a daily base (implied soil water measurements daily ? see above).
Include in table 3 and text page 5, row 25-28: what was assumed for the rooting zone of the Cerrado plot? $Z_f$
Statistical data analysis was done well with good uncertainties estimations.

Chapter 3.
With the tables and figures the results are consistent documented and described. Discuss more on page 8, row 20-25: why in table 5 results for Eta differed (because of different sites with different rainfall amounts, because of different methods e.g. Nobrega.
Page 8 row 30 following: discuss more the uncertainty of Cerrado vegetation rooting zone for the evapotranspiration calculation (depth of rooting zone you used is very sensitive for the residual in the water balance
Results for LUC to pasture are well in accordance to other studies, role of soil compaction should be discussed for this land use (see Nobrega 2017 and Meister et al. 2017).
Page 9 row 25 on: the chapter is misunderstanding comparing with Fig 4 (water table changes):
Row 26: water balance residuals represent not only soil water storage, as defined before (includes also deep infiltration – groundwater recharge !); authors argues that cerrado remove water from deeper soil horizons (that's right), but groundwater fluctuation is much higher in pasture and sugarcane (why?.

It will be fine, if table or figure with the soil water content over the measurement period can be added, than it can be seen how the unsaturated soil zone react different between land uses and cerrado. – In Fig. 4b 2015 there is a remarkable water table deepening, but high surplus of dS/dt – why?

Discussion chapter 3.4 (should be enlarged a little with):
Result that pasture and sugarcane increase surface runoff and decrease Eta are very common (not surprising); but for the residual (increased significantly) it must be discussed more carefull with differentiation in the role of deep infiltration (groundwater recharge relative high, interflow in the slope?, change of soil water content – see the measurements – not used for the discussion; infiltration rates between Cerrado and land use types are comparable? Compare with literature results.

Conclusions: page 12, row8: avoid term change in soil water storage (you mean the residual, much more than soil water storage (see above)
Page 12 row 11,12: no documentation that higher infiltration rates in wooded Cerrado compared to pasture and sugar cane – add this in the paper.

I agree, that such long term monitoring studies must be done, to compare it with often done pure water balance simulation studies. Point out in 4., what for important results in detail are valuable for further studies and water balance modelling for Cerrado Biome.

**In total: acceptance with mayor revision**

Please add in the references:

PROCESS-BASED MODELLING OF THE IMPACTS OF LAND USE CHANGE ON THE WATER BALANCE IN THE CERRADO BIOME (RIO DAS MORTES, BRAZIL)
Sarina Meister, Rodolfo L. B. Nobrega, Wolfgang Rieger, Ronja Wolf and Gerhard Gerold
ERDKUNDE 2017, 71/3, 241-266

Lamparter, G.; Nóbrega, R.. L. B.; Kovacs, K.; Amorim, R. S. and Gerold, G. (2016): Modelling hydrological impacts of agricultural expansion in two macro-catchments in Southern Amazonia, Brazil. In: Regional Environmental Change. https://doi.org/10.1007/s10113- 016-1015-2

---

## Referee Comment (RC3) · Anonymous Referee #2 · 20 Sep 2018

The authors assess the impact of four different land uses (bare soil, sugarcane, pasture and wooded cerrado) on water balance components (mainly runoff and actual evapotranspiration) monitored in experimental plots (5 m width, 20 m length and 9% slope). The paper is potentially interesting for the readers of Hess, however it requires major revisions. In sub-Section 2.2 ("Experimental setting and instrumentation"), please clearly describe the monitoring infrastructure (refer to Fig. 2) by adding information on soil moisture probes, monitoring wells which are described in other parts of the manuscript. Please add the thickness of the unsaturated zone (40 m) in Fig. 2. Not clear Kc-values for sugarcane in Table 2. Do they refer to monthly values during the growing season? How do you obtain field capacity and saturated hydraulic conductivity? Please specify the root zone depth for sugarcane and wooded cerrado. In

sub-Section 3.2 ("Groundwater table fluctuation"), the results are suspicious. The authors declare Ks=10-3 m d-1 for a sandy soil (very low if compared to tabulated values, see publications of Clapp and Hornberger, 1978; Schaap et al., 2001; Twarakavi et al., 2009 to mention a few). The main problem is the relationship between water storage change and water table fluctuation in Fig. 4. If Ks=10-3 m d-1, and hypothesizing full saturation of soil profile, we can apply the Darcy law with the unit gradient and water takes about 4000 days to bypass 40 m of soil and reach the water table. Please check if I am wrong. If I am approximately right, the relationship between soil water storage change and water table fluctuation should be influenced by a time lag. The main concern of this study is that the authors draw general conclusions on a small-scale, quite "homogeneous" test-site (experimental plots of 5 m width, 20 m length and 9% slope) by ignoring large-scale spatial heterogeneity of soil properties and topography. Hence this study can be considered as a preliminary survey for a more ambitious scientific investigation.

---

## Referee Comment (RC4) · Anonymous Referee #3 · 26 Sep 2018

This paper describes an experimental approach at the hillslope scale concerning the possible water partitioning trade-offs due to the LCLUC dynamics. I think this paper is relevant since it studies water flux in the Cerrado biome. The manuscript is interesting and well written. The results are original and represent an important contribution to the understanding of hydrological processes in the Cerrado. However, my main concern is that the problem statement is not clearly defined and that the field experimental description is not sufficient as it is. I think the paper is well written and the relevant literature cited, however it requires major revisions. Suggested corrections: All acronyms of the equations that are in the text should be in italic and the equation with the unit (equation 8). Page 3: The figure 1, is it to highlight Brazil? the most important is the monitoring, the details in the photos are too small. Please improve visibility. Page 3:

[Figure]

In sub-Section 2.2 Experimental setting and instrumentation-better describe the part of the monitoring. What was the measuring range for each equipment? What are the distances between the equipments? What is the size of the plot? What are the characteristics of the forest? (DBH, Height, Density) In table 1: Were used different equipment for monitoring the same variable in different plots (i.e. soil moisture)? What is the error of each piece of equipment? some tests were carried out to know the difference between devices? In figure 2, Why did not your measure soil moisture in the Bare soil? Please explain. Page 4: The paragraphs in lines 5 to 13, should be inserted in the sub-section 2.2. Page 5: How and when do you obtain soil field capacity and saturated hydraulic conductivity? Page 5, Table 3: remove this table, you can describe it in a paragraph. Page 6: In sub-section 2.4 and 2.5 describe more about these topics. Page 8, Results and discussion: I think you need to further describe the results and compare with other papers. The study would have been of more interest to readers if various published water flux models had been tested using the data. Page 12 Conclusions: The conclusion reads more like a summary of the paper.

―――――――――――――――――――――

---

## Author Comment (AC1) · 21 Oct 2018

We would like to thank the anonymous **referee 1** for the kind words in support of our manuscript and for the time spent reviewing our text. Here, we replied the referee's comments, which were highly insightful and enabled us to improve the quality of our manuscript. Note that the original referee's comments are identified as R1Cxx and written in **bold**, and the authors' responses are labeled as AR-R1Cxx. In addition, all comments are numbered (xx).

R1C1: The paper treat an important topic in the frame of LCLU for the Cerrado of Brazil. Until now only very few studies with experimental site data (see Oliveira, Nobrega) cover the Cerrado Biome in Brazil (most deals with Amazon

**rainforest). Problem statement is clear and well written.**

AR-R1C1: Thank you for recognizing the importance of the topic described by our manuscript. We hope to solve all the concerns and remarks found along the text to improve its comprehension and quality.

**R1C2: In the space of row 3 - 10 an outlook on the process of further Cerrado conversion should be added and why sugarcane in the study area will be important in this process of LUC.**

AR-R1C2: Thank you for the suggestions. We will add the following paragraph in the space of row 3 – 10 on page 2: "In the context of the Cerrado biome, the conversion of undisturbed vegetation and pasturelands to mechanized crop systems (e.g. sugarcane, corn and soybeans) indicates that this region in Brazil has a dynamic LCLU situation (Lapola et al., 2013). The sugarcane is the Brazilian backbone for energy security, as the ethanol production is the third most cultivated crop after soybeans and corn, reflecting the increasing demand for automotive fuels along the years (Leal et al., 2013;Rodrigues et al., 2018). Thus, the country is the world second largest ethanol producer and the Cerrado comprises the sugarcane expansion frontier due to the availability of water and pasturelands for the crop expansion (Bellezoni et al., 2018)."

R1C3: The aim of the study is well written (row 25-28). Experimental instrumentation is detailed described and adequate for the aim of longterm monitoring between the different land uses and Cerrado sensu stricto.

AR-R1C3: We appreciate your comment.
R1C4: In 2.2 following should be added for understanding the calculations later: Page 3: Time interval of soil water measurements (daily?) As basic information were ksat measurements done to understand the importance of infiltration to the groundwater of the Entisols?

AR-R1C4: The soil moisture was measured every 10 minutes; however, in this study we used daily averages as our time resolution in this study was daily. We will add the following phrase by the end of row 13 on page 3: "All instruments recorded data every 10 minutes, except the pressure transducers, which logged the groundwater table twice a day." We have ksat information of the study area: 102.279 mm h-1 (20 cm depth); 11.302 mm h-1 (50 cm depth); and 19.813 mm h-1 (100 cm depth). We will add this information in the revised version of the manuscript.

R1C5: Page 4 row 3: it seems better to define surface runoff as Qsur or Of (overland flow) instead of Q, because in most hydrological studies Q is defined as total discharge (see hydrological terms).

AR-R1C5: Thank you for the suggestion. We will change the abbreviation for surface runoff along the text, figures and tables to OF (overland flow).

R1C6: Evapotranspiration was calculated in the standard form on the base of Penman-Monteith (ETo). Water stress coefficient was calculated on a daily base (implied soil water measurements daily ? see above).

AR-R1C6: Yes, the water stress coefficient (Ks) was calculated using the daily soil moisture from the FDR probes. We clarify the instruments measurement interval previously (AR-R1C4).

HESSD
**R1C7: Include in table 3 and text page 5, row 25-28: what was assumed for the rooting zone of the Cerrado plot? Zf**

AR-R1C7: The method that was used to calculate the evapotranspiration for the Wooded Cerrado did not used the rooting zone depth as an input parameter as shown in Eq. 7. However, we will add the wooded Cerrado rooting zone depth in section 2.2 (row 25, page 3), when we give more details about the land uses considered in this study: "The soil root zone in the wooded Cerrado may reach up to 18 m (Rawitscher, 1948). However, most of the water used for plants' transpiration comes from the first layers (up to 7.5 m) (Oliveira et al., 2005;Canadell et al., 1996)".

**R1C8: Statistical data analysis was done well with good uncertainties estimations.**

AR-R1C8: We appreciate this comment.

R1C9: Chapter 3. With the tables and figures the results are consistent documented and described. Discuss more on page 8, row 20-25: why in table 5 results for Eta differed (because of different sites with different rainfall amounts, because of different methods e.g. Nobrega.

AR-R1C9: This is an important remark. We appreciated the recognition of our tables and figures consistency. We can observe significant different values among the listed studies due to the multiple locations and methods considered. Our main idea was to evidence this huge variability among the reference studies. Thus, we will add an extra piece of discussion on page 8 (row 23) to clarify the main reason for these differences and change the paragraph structure. "(...) (Table 5) due to the diverse rainfall patterns among the study sites and the different methods used to measure or estimate the evapotranspiration." **HESSD**
R1C10: Page 8 row 30 following: discuss more the uncertainty of Cerrado vegetation rooting zone for the evapotranspiration calculation (depth of rooting zone you used is very sensitive for the residual in the water balance.

AR-R1C10: Thank you for this important remark. Actually, the methods used for the evapotranspiration estimates in the wooded Cerrado did not use the root depth as input parameter (see Equation 7). However, we recognize that it is important to discuss about the root zone depth in this paragraph. Thus, we will add a piece of discussion stating about the root zone uncertainty in the Cerrado (page 8, row 33). "However, the root zone depth of an undisturbed vegetation such as the wooded Cerrado is uncertain and may vary according to the soil characteristics (Canadell et al., 1996). It may influence the plants' transpiration (Rawitscher, 1948;Oliveira et al., 2005), and consequently the water balance residual."

R1C11: Results for LUC to pasture are well in accordance to other studies, role of soil compaction should be discussed for this land use (see Nobrega 2017 and Meister et al. 2017).

AR-R1C11: Yes, it is true to say that soil compaction in pasturelands will affect the water balance results. Thus, we will change the text at this point (page 9, row 20) to consider the suggested studies, in order to enrich our discussion: "Additionally, the deforestation and agricultural land uses may increase soil compaction, as the LCLUC influence the hydrological patterns along the soil profile by evident modifications in the soil characteristics (bulk density, infiltration capacity, etc.) (Lamparter et al., 2016;Meister et al., 2017)."

R1C12: Page 9 row 25 on: the chapter is misunderstanding comparing with Fig 4 (water table changes): Row 26: water balance residuals represent not only

HESSD
soil water storage, as defined before (includes also deep infiltration – groundwater recharge !); authors argues that cerrado remove water from deeper soil horizons (that's right), but groundwater fluctuation is much higher in pasture and sugarcane (why?.

AR-R1C12: Thank you for this important question. This happens because more water reaches the water table in the pasture in comparison with the Wooded Cerrado. In the wooded Cerrado, the water uptake by the vegetation is higher due to the deeper and denser root system in comparison with pasture and sugarcane. In the sugarcane and pasture, the soil water that was not consumed by the plants and neither evaporated, continues to infiltrates along the unsaturated zone and the water uptake by the plants becomes unfeasible as the roots are shallow. Consequently, more water becomes available for deep infiltration, and this is evidenced by the significant water table fluctuation, which means that there is a higher groundwater recharge under the pasture in comparison with the wooded Cerrado (Fig 3f). This is explained in detail in section 3.2. Sections 3.1. and 3.2. will have their contents changed due to the new information added in Fig. 3.

R1C13: It will be fine, if table or figure with the soil water content over the measurement period can be added, than it can be seen how the unsaturated soil zone react different between land uses and cerrado. – In Fig. 4b 2015 there is a remarkable water table deepening, but high surplus of dS/dt – why?

AR-R1C13: We prepared a new figure containing the soil water content along the monitoring period and we attached in this revision as Fig. 3e. To answer the question about the water table deepening in 2014, we will add a piece of discussion in the text (page 10, row 3): "In the well located in the pasture, the water table fluctuated negatively along 2014 and 2015 due to the drought that happened in 2014 (Getirana, 2015). The water surplus of 2015-2016 happened due to the La Niña phenomena,
that raised the rainfall pattern after the long dry season of 2014-2015 (Kakatkar et al., 2018). Consequently, the water table raised along 2016-2017."

R1C14: Discussion chapter 3.4 (should be enlarged a little with): Result that pasture and sugarcane increase surface runoff and decrease Eta are very common (not surprising); but for the residual (increased significantly) it must be discussed more carefull with differentiation in the role of deep infiltration (groundwater recharge relative high, interflow in the slope?, change of soil water content – see the measurements – not used for the discussion; infiltration rates between Cerrado and land use types are comparable? Compare with literature results.

AR-R1C14: Thank you for the suggestion. We will add new citations in this section and a new paragraph explaining how significant is the water balance residual to the aquifer recharge (water table fluctuation).

R1C15: Conclusions: page 12, row8: avoid term change in soil water storage (you mean the residual, much more than soil water storage (see above) Page 12 row 11,12: no documentation that higher infiltration rates in wooded Cerrado compared to pasture and sugar cane – add this in the paper.

AR-R1C15: We will change the term "soil water storage " along the text to "water balance residual" and we will define this terminology on session 2.2 (page 3, row 30). "The water balance residual (dS/dt) includes subsurface flow, soil water storage, deep percolation and groundwater recharge." We will also comment along the text that due to the decreased runoff of the wooded Cerrado in comparison with pasture, sugarcane and bare soil, more water infiltrates through the soil and become readily available for the plants' consumption (page 9, row 24): "These reduced surface runoff
rates in the wooded Cerrado increase soil water infiltration in comparison with pasture, sugarcane and bare soil. Thus, higher infiltration rates increase plant water availability (Krishnaswamy et al., 2013)."

R1C16: I agree, that such long term monitoring studies must be done, to compare it with often done pure water balance simulation studies. Point out in 4., what for important results in detail are valuable for further studies and water balance modelling for Cerrado Biome. In total: acceptance with mayor revision

AR-R1C16: We acknowledge the insightful comments about our manuscript and the kind words in support of its publication. We will also add in conclusions that our results are useful for future research, both for discovery and modeling sciences.

R1C17: Please add in the references: PROCESS-BASED MODELLING OF THE IMPACTS OF LAND USE CHANGE ON THE WATER BALANCE IN THE CERRADO BIOME (RIO DAS MORTES, BRAZIL) Sarina Meister, Rodolfo L. B. Nobrega, Wolfgang Rieger, Ronja Wolf and Gerhard Gerold ERDKUNDE 2017, 71/3, 241-266 Lamparter, G.; Nóbrega, R.. L. B.; Kovacs, K.; Amorim, R. S. and Gerold, G. (2016): Modelling hydrological impacts of agricultural expansion in two macro-catchments in Southern Amazonia, Brazil. In: Regional Environmental Change. https://doi.org/10.1007/s10113- 016-1015-2

AR-R1C17: Thank you for the suggested references. As commented in AR-R1C11, we cited these studies to support the discussion of our results.

Figure caption

Figure 3: Water balance components for different LCLU rainfall, P (a); evapotranspiration, ET (b); surface runoff, Q (c); soil water storage, dS/dt (d); Right
axes present the cumulative sum of the variables represented by graphs (a), (b), (c) and (d). Soil moisture (e) for pasture, sugarcane and wooded Cerrado; and water table (f) depth of the monitoring wells located in site 1 (pasture) and site 2 (wooded Cerrado).

References

Bellezoni, R. A., Sharma, D., Villela, A. A., and Pereira Junior, A. O.: Water-energy-food nexus of sugarcane ethanol production in the state of Goiás, Brazil: An analysis with regional input-output matrix, Biomass and Bioenergy, 115, 108-119, 10.1016/j.biombioe.2018.04.017, 2018.

Canadell, J., Jackson, R. B., Ehleringer, J. B., Mooney, H. A., Sala, O. E., and Schulze, E.-D.: Maximum rooting depth of vegetation types at the global scale, Oecologia, 108, 583-595, 10.1007/bf00329030, 1996.

Getirana, A. C. V.: Extreme water deficit in Brazil detected from space, J Hydrometeorol, 17, 591-599, 10.1175/JHM-D-15-0096.1, 2015.

Kakatkar, R., Gnanaseelan, C., Deepa, J. S., Chowdary, J., and Parekh, A.: Role of ocean-atmosphere interactions in modulating the 2016 La Niña like pattern over the tropical Pacific, Dynamics of Atmospheres and Oceans, 83, 100-110, 10.1016/j.dynatmoce.2018.07.003, 2018.

Krishnaswamy, J., Bonell, M., Venkatesh, B., Purandara, B. K., Rakesh, K. N., Lele, S., Kiran, M. C., Reddy, V., and Badiger, S.: The groundwater recharge response and hydrologic services of tropical humid forest ecosystems to use and reforestation: Support for the "infiltration-evapotranspiration trade-off hypothesis", J Hydrol, 498, 191-209, 10.1016/j.jhydrol.2013.06.034, 2013.

HESSD
Lamparter, G., Nobrega, R. L. B., Kovacs, K., Amorim, R. S., and Gerold, G.: Modelling hydrological impacts of agricultural expansion in two macro-catchments in Southern Amazonia, Brazil, Regional Environmental Change, 18, 91-103, 10.1007/s10113-016-1015-2, 2016.

Lapola, D. M., Martinelli, L. A., Peres, C. A., Ometto, J. P. H. B., Ferreira, M. E., Nobre, C. A., Aguiar, A. P. D., Bustamante, M. M. C., Cardoso, M. F., Costa, M. H., Joly, C. A., Leite, C. C., Moutinho, P., Sampaio, G., Strassburg, B. B. N., and Vieira, I. C. G.: Pervasive transition of the Brazilian land-use system, Nat Clim Change, 4, 27-35, 10.1038/nclimate2056, 2013.

Leal, M. R. L. V., Horta Nogueira, L. A., and Cortez, L. A. B.: Land demand for ethanol production, Applied Energy, 102, 266-271, 10.1016/j.apenergy.2012.09.037, 2013.

Meister, S., Nobrega, R. L. B., Rieger, W., Wolf, R., and Gerold, G.: Process-based modelling of the impacts of land use change on the water balance in the Cerrado Biome (Rio das Mortes, Brazil), Erdkunde, 71, 241-266, 10.3112/erdkunde.2017.03.06, 2017.

Oliveira, R. S., Bezerra, L., Davidson, E. A., Pinto, F., Klink, C. A., Nepstad, D. C., and Moreira, A.: Deep root function in soil water dynamics in cerrado savannas of central Brazil, Functional Ecology, 19, 574-581, 10.1111/j.1365-2435.2005.01003.x, 2005.

Rawitscher, F.: The Water Economy of the Vegetation of the 'Campos Cerrados' in Southern Brazil, Journal of Ecology, 36, 237-268, 10.2307/2256669, 1948.

Rodrigues, N., Losekann, L., and Silveira Filho, G.: Demand of automotive fuels in Brazil: Underlying energy demand trend and asymmetric price response, Energy Economics, 74, 644-655, 10.1016/j.eneco.2018.07.005, 2018.

---

## Author Comment (AC2) · 21 Oct 2018

We would like to thank the anonymous **referee 2** for the kind words in support of our manuscript and for the time spent reviewing our text. Here, we replied the referee's comments, which were highly insightful and enabled us to improve the quality of our manuscript. Note that the original referee's comments are identified as R2Cxx and written in **bold**, and the authors' responses are labeled as AR-R2Cxx. In addition, all comments are numbered (xx).

**R2C1: The authors assess the impact of four different land uses (bare soil, sugarcane, pasture and wooded cerrado) on water balance components (mainly runoff and actual evapotranspiration) monitored in experimental plots (5 m**

[Figure]

**width, 20 m length and 9% slope). The paper is potentially interesting for the readers of Hess, however it requires major revisions.**

AR-R2C1: We appreciate the reviewer's comments and suggestions and we recognize that our manuscript requires more detailed descriptions of some topics listed by Reviewer 2. We hope to solve the problems found along the text to improve its comprehension and quality. In addition, it is important to remark that this is a site-specific study, and the discussion and conclusions stated in here are based on field observed data, without any pretention to generalize to a larger area. Our scope was to investigate the hydrological process in detail by adopting the hillslope scale as the design criteria of the experimental setup. Thus, our results will help further studies supporting them with 5-year experimental data that confirms the significant influence of the land use in the water partitioning in a subtropical region.

**R2C2: In sub-Section 2.2 ("Experimental setting and instrumentation"), please clearly describe the monitoring infrastructure (refer to Fig. 2) by adding information on soil moisture probes, monitoring wells which are described in other parts of the manuscript. Please add the thickness of the unsaturated zone (40 m) in Fig. 2.**

AR-R2C2: Thank you for the suggestion. We added the thickness of the unsaturated zone in Fig. 2 (see Fig. 2) and additional information will be added in section 2.2 to better describe the monitoring wells and soil moisture probes.

**R2C3: Not clear Kc-values for sugarcane in Table 2. Do they refer to monthly values during the growing season? How do you obtain field capacity and saturated hydraulic conductivity? Please specify the root zone depth for sugarcane and wooded cerrado.**

AR-R2C3: Thank you for requesting this information. The Kc value vary along the year. We will specify in a new version of Table 2 (see the supplement to this comment) the months that the different Kc values are referring to. We obtained the soil field capacity and saturated hydraulic conductivity using Büchner funnels and Richards extraction chambers (information to be added in the revised manuscript). The root zone depth for sugarcane was specified on Table 2 (see the second column). However, the root zone depth for the wooded Cerrado will be specified in the revised version in section 2.2 (row 25, page 3): "The soil root zone in the wooded Cerrado may reach up to 18 m (Rawitscher, 1948). However, most of the water used for plants' transpiration comes from the first layers (up to 7.5 m) (Oliveira et al., 2005;Canadell et al., 1996)".

**R2C4: In sub-Section 3.2 ("Groundwater table fluctuation"), the results are suspicious. The authors declare Ks=10-3 m d-1 for a sandy soil (very low if compared to tabulated values, see publications of Clapp and Hornberger, 1978; Schaap et al., 2001; Twarakavi et al., 2009 to mention a few). The main problem is the relationship between water storage change and water table fluctuation in Fig. 4. If Ks=10-3 m d-1, and hypothesizing full saturation of soil profile, we can apply the Darcy law with the unit gradient and water takes about 4000 days to bypass 40 m of soil and reach the water table. Please check if I am wrong. If I am approximately right, the relationship between soil water storage change and water table fluctuation should be influenced by a time lag.**

AR-R2C4: Thank you for the remark. Our idea was to verify the aquifer hydraulic conductivity in both wells (pasture and wooded Cerrado) to be sure that the wells conditions were the same and consequently comparable in terms of water table fluctuation. The hydraulic conductivity mentioned in the section 3.2 was obtained by a slug test. Thus, it is not referring to the hydraulic conductivity for the sandy soil from the unsaturated zone. In fact, it is referring to the aquifer hydraulic conductivity, which

is a sandstone. The soil hydraulic conductivity in the upper layers of the soil (up to 1 m depth) is around $10^{-1}$ m d$^{-1}$. Thus, the soil porosity, and consequently, the soil hydraulic conductivity may vary along the unsaturated zone.

**R2C5: The main concern of this study is that the authors draw general conclusions on a small-scale, quite "homogeneous" test-site (experimental plots of 5 m width, 20 m length and 9% slope) by ignoring large-scale spatial heterogeneity of soil properties and topography. Hence this study can be considered as a preliminary survey for a more ambitious scientific investigation.**

AR-R2C5: We appreciate your concern about the scale factor. There is a need to understand the water partitioning in the Cerrado region at multiple scales (Oliveira et al., 2014). In this study, our priority was to investigate the water balance at the hillslope scale in multiple land uses. The choice of the evaluated LCLU was based on the current LCLUC dynamics in the Cerrado described in the introduction and in the comment AR-R1C2. In the future, the findings of the present study can be used for data validation in spatial scale studies and be part of a multi-scale approach to evaluate the water balance in the Cerrado.

Figure caption

Figure 2: Hydrological monitoring performed on the four treatments: (1) relative humidity and air temperature probes at 2 m (site 1) and 11 m (site 2); (2) rainfall gauges; (3) soil moisture sensors; (4) solar radiation sensor; (5) wind speed and direction (anemometers) at 2 m (site 1) and at 11 m (site 2); (6) surface runoff collectors; (7) monitoring wells equipped with water table pressure transducers; (8) net radiation sensor; (9) soil heat flux plate.

References

Allen, R. G., Pereira, L. S., Raes, D., and Smith, M.: Crop evapotranspiration - Guidelines for computing crop water requirements - FAO Irrigation and drainage paper 56, FAO - Food and Agriculture Organization of the United Nations, Rome, 1998.

Canadell, J., Jackson, R. B., Ehleringer, J. B., Mooney, H. A., Sala, O. E., and Schulze, E.-D.: Maximum rooting depth of vegetation types at the global scale, Oecologia, 108, 583-595, 10.1007/bf00329030, 1996.

Doorenbos, J., Pruitt, W. O., Aboukhaled, A., Food, and Nations, A. O. o. t. U.: Guidelines for Predicting Crop Water Requirements, Food and Agriculture Organization of the United Nations, 1975.

Doorenbos, J., Kassam, A. H., and Bentvelsen, C. I. M.: Yield response to water, Food and Agriculture Organization of the United Nations, 1979.

Oliveira, P. T. S.: Balanço hídrico e erosão do solo em mata nativa do bioma Cerrado, PhD, Departamento de Hidráulica e Saneamento - Escola de Engenharia de São Carlos, Universidade de São Paulo, São Carlos, SP, 2014.

Oliveira, P. T. S., Nearing, M. A., Moran, M. S., Goodrich, D. C., Wendland, E., and Gupta, H. V.: Trends in water balance components across the Brazilian Cerrado, Water Resour Res, 50, 7100-7114, 10.1002/2013WR015202, 2014.

Oliveira, R. S., Bezerra, L., Davidson, E. A., Pinto, F., Klink, C. A., Nepstad, D. C., and Moreira, A.: Deep root function in soil water dynamics in cerrado savannas of central Brazil, Functional Ecology, 19, 574-581, 10.1111/j.1365-2435.2005.01003.x, 2005.

[Figure]

Rawitscher, F.: The Water Economy of the Vegetation of the 'Campos Cerrados' in Southern Brazil, Journal of Ecology, 36, 237-268, 10.2307/2256669, 1948.

Please also note the supplement to this comment:
https://www.hydrol-earth-syst-sci-discuss.net/hess-2018-415/hess-2018-415-AC2-supplement.pdf

Agricultural land – Site 1

Natural conditions – Site 2

Pasture | Sugarcane | Bare soil | Wooded Cerrado

40 m

Sandy soil

Aquifer

**Fig. 1.** Figure 2 (see caption above)

**Supplement:**

**Table 2: Variables used to calculate evapotranspiration for sugarcane and pasture.**

| Variable | Sugarcane | Pasture | Source |
|---|---|---|---|
| $Z_f$ (m) | 1.2 – 2.0 | 0.5 – 1.5 | Allen et al. (1998) |
| p | 0.65 | 0.60 | |
| $\theta_{fc}$ | 0.14 | 0.14 | Values obtained from laboratory essays (Oliveira, |
| $\theta_{wp}$ | 0.09 | 0.09 | 2014) |
| Kc | Plant (1): 0.50[a]; 0.80[b]; 095[c]; 1.10[d]; 1.18[e]; 0.92[f]; 0.68[g]
 Ratoon (2): 0.55[a]; 0.80[b]; 090[c]; 1.00[d]; 1.05[e]; 0.80[f]; 0.60[g] | 0.75 (3) | (1) Doorenbos et al. (1975)
 (2) Doorenbos et al. (1979)
 (3) Allen et al. (1998) |

Approximate sugarcane age (days): a (0-30); b (30-60); c (60-75); d (75-120); e (120-300); f (300-330), g (330-360).

| Variable | Sugarcane | Pasture | Source |
|---|---|---|---|
| $Z_f$ (m) | 1.2 – 2.0 | 0.5 – 1.5 | Allen et al. (1998) |
| p | 0.65 | 0.60 | |
| $\theta_{fc}$ | 0.14 | | |
| $\theta_{wp}$ | 0.09 | | |
| Kc | Plant (1): 0.55[a]; 0.80[b]; 090[c]; 1.00[d]; 1.05[e]; 0.80[f]; 0.60[g] | | |

---

## Author Comment (AC3) · 21 Oct 2018

We would like to thank the anonymous **referee 3** for the kind words in support of our manuscript and for the time spent reviewing our text. Here, we replied the referee's comments, which were highly insightful and enabled us to improve the quality of our manuscript. Note that the original referee's comments are identified as R3Cxx and written in **bold**, and the authors' responses are labeled as AR-R3Cxx. In addition, all comments are numbered (xx).

**R3C1: This paper describes an experimental approach at the hillslope scale concerning the possible water partitioning trade-offs due to the LCLUC dynamics. I think this paper is relevant since it studies water flux in the Cerrado**

[Figure]

**biome. The manuscript is interesting and well written. The results are original and represent an important contribution to the understanding of hydrological processes in the Cerrado. However, my main concern is that the problem statement is not clearly defined and that the field experimental description is not sufficient as it is. I think the paper is well written and the relevant literature cited, however it requires major revisions.**

AR-R3C1: We appreciate the reviewer feedback about our manuscript and the time spent during the reading and further revision. Here, we express our accordance with the relevant given suggestions. We also added the information that will be included in a revised version of the manuscript. We will deeply revise the problem statement, which is mainly based on the lack of field studies in the tropics considering different land uses and an undisturbed condition (Cerrado). We recognize that it could contain more arguments involving the need of field observed data for both modeling and discovery sciences.

**Suggested corrections:**

**R3C2: All acronyms of the equations that are in the text should be in italic and the equation with the unit (equation 8).**

AR-R3C2: Thank you for the correction. We will change this along the text according to the instructions.

**R3C3: Page 3: The figure 1, is it to highlight Brazil? the most important is the monitoring, the details in the photos are too small. Please improve visibility.**

AR-R3C3: Thank you for your suggestion. We wanted to highlight the study area context by printing the country larger than the other information. However, we

recognized that it is more important to show the reader more information about the study site itself. Thus, we made a new Fig.1 (see below) considering this important suggestion.

**R3C4: Page 3: In sub-Section 2.2 Experimental setting and instrumentation-better describe the part of the monitoring. What was the measuring range for each equipment? What are the distances between the equipments? What is the size of the plot? What are the characteristics of the forest? (DBH, Height, Density)**

AR-R3C4: Thank you for these important questions. Concerning the measuring range of each equipment, we will add this information on Table 1 (see the supplement to this comment). The distance between sites 1 and 2 are 1.7 km. Site 1 has 9 plots placed side by side (approximately 2.5 meters of distance between each plot) (see Figure 1). Also in site 1, there is a meteorological station that concentrates almost all sensors placed at that site, except for the soil moisture probes (we have one placed inside the first sugarcane plot, and 20 m to the left, we have another one placed inside the first pasture plot). Site 2 has 3 plots inside a tropical woodland (wooded Cerrado) and due to the tree density and topography, the plots are approximately 5 to 10 meters distant from each other. Approximately 50 m to the north from the plots, we have a meteorological tower (11 m height) containing all the sensors placed at site 2. Concerning the forest characteristics, the wooded Cerrado area used in this study has 15522 individuals per hectare, the height of most of the trees is about 8 m, and the diameters (DBH) are predominantly between 3 and 7 cm (Reys et al., 2013). We will add an extra paragraph in section 2.2 in order to better describe the instruments' positions and the forest characteristics.

**R3C5: In table 1: Were used different equipment for monitoring the same variable in different plots (i.e. soil moisture)? What is the error of each piece of**

**equipment? some tests were carried out to know the difference between devices?**

AR-R3C5: Thank you for this important remark. We used the same model of soil moisture probes in the different probes (see the supplement to this comment). We added the maximum error of each piece of equipment in Table 1 (see the supplement to this comment). The soil moisture probes had their first use in our study sites and they were all previously calibrated with soil samples from our study sites.

**R3C6: In figure 2, Why did not your measure soil moisture in the Bare soil? Please explain.**

AR-R3C6: Thank you for the question. We had not enough ports in the datalogger to connect another soil moisture probe to monitor the bare soil plot. In addition, we did not have a piece of equipment available to perform such monitoring.

**R3C7: Page 4: The paragraphs in lines 5 to 13, should be inserted in the sub-section 2.2.**

AR-R3C7: We agree that this information suits on section 2.2. However, we added these paragraphs in section 2.3 (water balance components) because each of them is describing how we obtained each of the water balance components: Page 4, lines 5-6 describe how we monitor the rainfall; Page 4, lines 7-9 describe how we obtained the overland flow and how we calculated the runoff coefficient; Page 4, lines 10-12 describe how we estimated the reference evapotranspiration. Thus, we argue that removing these paragraphs from section 2.3, we may lose the sequence of the text. You can see that just after line 13 (page 4), we present Equation 2, which is also referred in line 13. Additionally, these paragraphs define how we obtained the input variables used in Equation. 1.

**R3C8: Page 5: How and when do you obtain soil field capacity and saturated hydraulic conductivity?**

AR-R3C8: We obtained the soil field capacity and saturated hydraulic conductivity using Büchner funnels and Richards extraction chambers. These tests were performed in the beginning of the experiment (2012) (Oliveira, 2014). We collected the samples with undisturbed structure in volumetric rings at depths of 20, 50 and 100 cm.

**R3C9: Page 5, Table 3: remove this table, you can describe it in a paragraph.**

AR-R3C9: Thank you for the suggestion. We will remove Table 3 and a new paragraph will be added after line 22 (page 5): "The Priestley and Taylor coefficients ($\alpha$) calculated for a wooded Cerrado area close to the study site (Cabral et al., 2015) differed according to the season: 1.09 for Summer (December – March); 1.00 for Fall (March – June); 0.77 for Winter (June – September); and 0.98 for Spring (September – December).".

**R3C10: Page 6: In sub-section 2.4 and 2.5 describe more about these topics.**

AR-R3C10: Thank you for the suggestion. We completed these 2 paraghaphs with additional information. We will modify the paragraphs as reported below:

Section 2.4: Groundwater table fluctuation "The water table was registered twice a day (at 6 am and 6 pm) using pressure transducers (Diver, Schlumberger) placed inside two monitoring wells (well 1 located in the pasture area; and well 2 located inside the wooded Cerrado area). In the study site, both wells presented similar hydraulic conductivity according to the slug test (Bouwer and Rice, 1976) previously performed. We evaluated the aquifer hydraulic conductivity from both wells in order to

validate the water table comparison among each other, as whether the aquifer condition in the wells were different, the such comparison would not be fair. Both wells reach the water table at approximately 40 m depth in an unconfined sandstone formation (Botucatu formation), which belongs to São Bento Group of Mesozoic age. In addition, the soils above the aquifer that appears thought the unsaturated zone are Cenozoic sediments weathered from the sandstone (Wendland et al., 2007)."

Section 2.5: Data analysis "The normality assumption was tested using the Shapiro-Wilk test using a 95% confidence interval for rainfall, evapotranspiration, surface runoff and soil water storage datasets. The one-way analysis of variance (ANOVA) was applied to test the null and alternative hypothesis, that is, equality of surface runoff, evapotranspiration and soil water storage distribution functions between the four treatments (LCLU) versus the difference in distribution functions between at least two treatments. Additionally, the multiple comparisons between treatments were performed using the Tukey test (Montgomery, 2008). The rainfall, evapotranspiration, surface runoff, water balance residual and soil moisture graphs were plotted using a daily basis timescale. The groundwater table fluctuation was plotted using a monthly timescale due to the noise typically found in this kind of measurement. In order to present the order of magnitude along the years, the data was also resumed annually in tables and figures."

**R3C11: Page 8, Results and discussion: I think you need to further describe the results and compare with other papers. The study would have been of more interest to readers if various published water flux models had been tested using the data.**

AR-R3C11: Thank you for your suggestion. As asked by other reviewers too, we will improve the results' discussion by contrasting our outcomes with other studies in the revised manuscript to be submitted. Concerning the testing of water flux models, it

was not part of our scope to test models using our data. We consider that water flux model testing with the data presented along this study should be part of a new study. Thus, we may add it as a recommendation along the discussion and also in the concluding remarks. We believe that the main contribution of our study is the long-term monitoring at the hillslope scale under subtropical conditions. Such kind of data is a resource for both discovery and modeling sciences . Additionally, we could draw significant conclusions by the comparisons of the contrasting land uses considered in this study.

**R3C12: Page 12 Conclusions: The conclusion reads more like a summary of the paper.**

AR-R3C12: We recognize this aspect. Along the revision, we will add substantial information in the results and discussion session. Thus, in the revised version, we will add other assumptions discussed thought the manuscript. The fact that we summarize the manuscript in the first paragraph of the conclusion is due to the need of remember the reader about the context of our study. In addition, some readers go straight to the conclusions in a first read of a paper and when we give at least a brief description of the study before giving the conclusions, we improve the comprehension of our scientific contributions.

Figure caption

Figure 1: Location of study sites, Cerrado biome borders, Guarani Aquifer System (GAS) outcrop zone distribution in Brazil, and experimental design, where site 1 contains the plots with agricultural land uses (pasture, sugarcane and bare soil) and site 2 contains the plots with wooded Cerrado.

References

Bouwer, H., and Rice, R. C.: A slug test for determining hydraulic conductivity of unconfined aquifers with completely or partially penetrating wells, Water Resour Res, 12, 423-428, 10.1029/WR012i003p00423, 1976.

Cabral, O. M. R., da Rocha, H. R., Gash, J. H., Freitas, H. C., and Ligo, M. A. V.: Water and energy fluxes from a woodland savanna (cerrado) in southeast Brazil, J Hydrol: Regional Studies, 4, 22-40, 10.1016/j.ejrh.2015.04.010, 2015.

Montgomery, D. C.: Design and Analysis of Experiments, John Wiley  Sons, 752 pp., 2008.

Oliveira, P. T. S.: Balanço hídrico e erosão do solo em mata nativa do bioma Cerrado, PhD, Departamento de Hidráulica e Saneamento - Escola de Engenharia de São Carlos, Universidade de São Paulo, São Carlos, SP, 2014.

Reys, P., Camargo, M. G. G., Grombone-Guarantini, M. T., Teixeira, A. P., Assis, M. A., and Morellato, L. P. C.: Estrutura e composição florística de um Cerrado sensu stricto e sua importância para propostas de restauração ecológica, Hoehnea, 40, 449-464, 2013.

Wendland, E., Barreto, C., and Gomes, L. H.: Water balance in the Guarani Aquifer outcrop zone based on hydrogeologic monitoring, J Hydrol, 342, 261-269, 10.1016/j.jhydrol.2007.05.033, 2007.

Please also note the supplement to this comment:
https://www.hydrol-earth-syst-sci-discuss.net/hess-2018-415/hess-2018-415-AC3-supplement.pdf

[Figure]

Fig. 1. Figure 1 (see caption above)

**Supplement:**

**Table 1: Monitoring site instrumentation characteristics.**

| Variable | Sensor | Height or depth (m) | Measurement range | Maximum error | Site no. |
|---|---|---|---|---|---|
| Temperature (°C) and relative humidity (%) | HMP45C | 2.0 | -39.2 to +60°C and 0.8% to 100% | ±0.5°C and ±3% | 1 |
| | HC2S3 | 9.5 and 11.0 | | ±0.1°C and ±0.8% | 2 |
| Rainfall (mm) | Hydrological Services TB4 | 1.5 and 11.0 | 0 to 700 mm h$^{-1}$ | ±3% | 1 and 2 |
| Atmospheric pressure (mbar) | Vaisala CS106 | 1.0 | 500 to 1100 mBar | ±1.5 mBar | 1 |
| Wind direction (°) and velocity (m.s$^{-1}$) | Young 05103 | 2.0 and 11.0 | 0 to 360° and 0 to 100 m s$^{-1}$ | ±3° and ±0.3 m s$^{-1}$ | 1 and 2 |
| Solar radiation (MJ.m$^{-2}$) | Kipp & Zonen CMP3 | 2.0 | 0 to 2000 W m$^{-2}$ | ±5% | 1 |
| Soil moisture (%) | FDR Environscan Sentek | -0.3; -0.6 and -0.9 | 0 to ~65% | ±3% | 1* |
| | | -0.1; -0.5; -0.7; -1.0 and -1.5 | | | 2 |
| Net solar radiation (W.m$^{-2}$) | Kipp & Zonen NR-LITE2 | 11.0 | ±2000 W m$^{-2}$ | ±5% | 2 |
| Soil heat flux (W.m$^{-2}$) | Hukseflux HFP01 | -0.1 | ±2000 W m$^{-2}$ | -15% to +5% | 2 |
| Water table (groundwater) (m) | Diver Schlumberger | -40.0 and -39.2 | 0 to 10 m | ± 2.5 cm | 1 and 2 |

*At site 1, the bare soil did not have soil moisture probes.

---

## Author Comment (AC4) · 21 Oct 2018

Please, find the responses to referee 1 in the "RC2" reply ("AC1" in the interactive discussion).
* * *

---

## Author Comment (AC5) · 30 Oct 2018

We detected an error in Figure 3 and we uploaded a new version.

Figure 3: Water balance components for different LCLU rainfall, P (a); evapotranspiration, ET (b); surface runoff, OF (c); soil water storage, dS/dt (d); Right axes present the cumulative sum of the variables represented by graphs (a), (b), (c) and (d). Soil moisture (e) for pasture, sugarcane and wooded Cerrado; and water table (f) depth of the monitoring wells located in site 1 (pasture) and site 2 (wooded Cerrado).

[Figure]

[Figure]

**Fig. 1.** Figure 3 (see caption above)

---

## Author Response (AR1)

**To: Dr Martijn Westhoff**
Associate Editor
*HESS*

**From: Dr Jamil A. A. Anache**
Corresponding author
*EESC-USP*

December 7[th], 2018

Dear Dr. Westhoff,

Re: Manuscript reference HESS-2018-415

Please find attached a revised version of our manuscript entitled "**Hydrological trade-offs due to different land covers and land uses in the Brazilian Cerrado**" by Jamil A. A. Anache, Lívia M. P. Rosalem, Edson Wendland, Cristian Youlton and Paulo T.S. Oliveira for publication in Hydrology and Earth System Sciences. We are also sending the items: changes marked, and response to reviewers' comments (revision notes).

Note that in "Revision Notes" the original editor and reviewer comments are in **bold** and author responses are in normal text throughout. Pages and lines indicated in the "Revision Notes" refer to the "Revised Manuscript". In "Changes Marked" we highlight the changes made in the text that were indicated in the comments from the editor and reviewers.

We would like to thank the editor and reviewers for their kind words in support of our manuscript and for their time spent reviewing our text. The manuscript has been revised in accordance to their comments, which were highly insightful and enabled us to improve the quality of our manuscript.

We hope that the revisions in the manuscript and our accompanying responses will now meet the requirements for publication in Hydrology and Earth System Sciences.

Thank you again for your consideration.

Yours sincerely,

Jamil A. A. Anache
*Post-doctoral research fellow at the Computational Hydraulics Laboratory*
*São Carlos School of Engineering, University of São Paulo*

**Revision Notes**

Editor's comments

**ECC1: Dear Authors, you have received three reviews, which all provided clear feedback on how to improve the manuscript. Although all questions raised by the reviewers should be adressed, I consider two of them as rather important ones:**

**ECC2: Referee 1 correctly mentioned that dS/dt is a water residual (including storage, recharge and deep infiltration). I think it is important to make that clear from the beginning.**

AR-ECC2: Thank you for this important remark concerning the dS/dt definition along the text. We agree that it is essential to make it clear that it represents the water balance residual, which includes soil water storage, deep infiltration and subsurface flow. Thus, we standardized all the dS/dt mentions, and we referred it as the water balance residual.

**ECC3: Referee 2 mentioned the time lag between precipitation and GW recharge. Even if the hydraulic conductivity of the unsaturated soil is not 10^-3 but 10^-1, the travel time for a rain drop to reach the groundwater will still be in the order of 1 year. This means that the groundwater fluctuations are completely disconnected to what happens on the surface. The GW fluctuations also reflect a much larger area than the plot scale. So maybe, the GW fluctuations should be left out of the analysis completely. Instead the focus should be on the top soil (up to the rooting depth?).**

AR-ECC3: Thank you for this remark. The idea of comparing the water balance results with the groundwater table fluctuations from wells located in wooded Cerrado and pasture land covers was to evaluate if the surface hydrological processes would reflect in the groundwater level. We inserted this approach because in our previous studies we have found the influence of land cover and land use (LCLU) in the water table fluctuations (Lucas and Wendland, 2015; Lucas et al., 2015; Oliveira et al., 2017; Leite et al., 2018) (page 11, line 8 – line 18). Furthermore, it is important to note that the unsaturated zone thickness and material, topography and aquifer hydraulic conductivity were similar in both wells, and their locations represent homogenous surface conditions, as they are not located in the edges of the LCLU that they represent. We added brief discussion in the chapter 3.2 to address the limitations of the assumptions regarding the groundwater table fluctuation in comparison with the water balance. We also removed Figure 4, as Figure 3F reflects the same phenomena with less evidence to it, as the effects of LCLU on groundwater level tend to be non-linear and difficult to analyze as they result from complex interactions between LCLU and hydrological processes (Han et al., 2017). Additionally, by removing completely the groundwater fluctuations from our manuscript, we may loss an interesting hypothesis for further investigations in this subject, and not mentioning the fact that the study site is located above the outcrop zone of a major aquifer in South America, the Guarani Aquifer System (GAS), thus, the groundwater plays an important role on the site contextualization.

It is also important to make clear that the wells show the groundwater table fluctuations in the respective areas (Cerrado and pasture covers) and not for a small plot as suggested by the Reviewer 2. We have inserted a brief sentence to make it clear to the readers. (page 11, line 26 – line 29).

**ECC4: On top of these two comments, I have another comment about the error calculations: When calculating the water balance residuals, a lot of uncertainty lies in the evaporation estimates. The error is not only coming from random noise in the measurements, but also from the used formulation (e.g. Penmann-Monteith vs. Priestley-Taylor). Besides that, the fact that you come up with a lower error estimate for the Cerrado site, seems to arise from the fact that in PM more parameters are required, which all sum-up in the error calculation (Eq. 11) (unless you have a clear idea on the error in the Priestly and Taylor coefficient).**

AR-ECC4: Thanks for pointing this out. We performed a new uncertainty assessment considering the Priestley-Taylor coefficient estimation errors and we found a standard error with similar order of magnitude of the PM method. This leaded to a more homogeneous and fair comparison between the different LCLU water balance outcomes. We updated Table 5 and Figure 4, which contains the results from the data uncertainty chapter.

**ECC5: Please make also sure that units for all equations are correct (this is not the case for (at least) Eq. 3). Also be aware that multi-letter variables should be avoided (see the mathematical requirements at https://www.hydrology-and-earth-system-sciences.net/for_authors/manuscript_preparation.html). With kind regards, Martijn Westhoff.**

AR-ECC5: We appreciate the advice and corrections concerning the mathematical requirements. We double-check all equations, units and variables through the text to ensure the compliance of the journal rules. In addition to the mathematical requirements, we made the datasets available in a data repository (HydroShare), following the journal recommendations (page 14, line 16 – line 18).

Reviewer 1 comments

We would like to thank the anonymous Referee 1 for the kind words in support of our manuscript and for the time spent reviewing our text. Here, we replied the referee's comments, which were highly insightful and enabled us to improve the quality of our manuscript. Note that the original referee's comments are identified as R1Cxx and written in **bold**, and the authors' responses are labeled as AR-R1Cxx. In addition, all comments are numbered (xx).

**R1C1: The paper treat an important topic in the frame of LCLU for the Cerrado of Brazil. Until now only very few studies with experimental site data (see Oliveira, Nobrega) cover the Cerrado Biome in Brazil (most deals with Amazon rainforest). Problem statement is clear and well written.**

AR-R1C1: Thank you for recognizing the importance of the topic described by our manuscript. We replied and solved, when possible, all the concerns and remarks found along the text to improve its comprehension and quality.

**R1C2: In the space of row 3 – 10 an outlook on the process of further Cerrado conversion should be added and why sugarcane in the study area will be important in this process of LUC.**

AR-R1C2: Thank you for the suggestions. We added the following paragraph on page 2 (line 11-line 16): "In the context of the Cerrado biome, the conversion of undisturbed vegetation and pasturelands to mechanized crop systems (e.g. sugarcane, corn and soybeans) indicates that this region in Brazil has a dynamic LCLU situation (Lapola et al., 2013). The sugarcane is the Brazilian backbone for energy security, as the ethanol production is the third most cultivated crop after soybeans and corn, reflecting the increasing demand for automotive fuels along the years (Leal et al., 2013; Rodrigues et al., 2018). Thus, the country is the world second largest ethanol producer and the Cerrado comprises the sugarcane expansion frontier due to the availability of water and pasturelands for the crop expansion (Bellezoni et al., 2018).."

**R1C3: The aim of the study is well written (row 25-28). Experimental instrumentation is detailed described and adequate for the aim of longterm monitoring between the different land uses and Cerrado sensu stricto.**

AR-R1C3: We appreciate your comment.

**R1C4: In 2.2 following should be added for understanding the calculations later: Page 3: Time interval of soil water measurements (daily?) As basic information were ksat measurements done to understand the importance of infiltration to the groundwater of the Entisols?**

AR-R1C4: The soil moisture was measured every 10 minutes; however, we used daily averages, as our temporal resolution in the present study was daily. We added the following sentence on page 3 (line 17

– line 18): "All instruments recorded data every 10 minutes, except the pressure transducers, which logged the groundwater table twice a day." We have ksat information of the study area, added on section 2.1 (page 3, line 9 – line 10): 102.279 mm h-1 (20 cm depth); 11.302 mm h-1 (50 cm depth); and 19.813 mm h-1 (100 cm depth). We added this information in the revised version of the manuscript.

**R1C5: Page 4 row 3: it seems better to define surface runoff as Qsur or Of (overland flow) instead of Q, because in most hydrological studies Q is defined as total discharge (see hydrological terms).**

AR-R1C5: Thank you for the suggestion. We changed the abbreviation for surface runoff along the text, figures and tables to $O_F$ (overland flow).

**R1C6: Evapotranspiration was calculated in the standard form on the base of Penman-Monteith (ETo). Water stress coefficient was calculated on a daily base (implied soil water measurements daily ? see above).**

AR-R1C6: Yes, the water stress coefficient ($K_S$) was calculated using the daily soil moisture from the FDR probes. We clarify the instruments measurement interval previously (AR-R1C4). It is also mentioned on page 5, line 11 that the $K_S$ was calculated at a daily basis.

**R1C7: Include in table 3 and text page 5, row 25-28: what was assumed for the rooting zone of the Cerrado plot? Zf**

AR-R1C7: The method that was used to calculate the evapotranspiration for the Wooded Cerrado did not used the rooting zone depth as an input parameter as shown in Eq. 7. However, we added the wooded Cerrado rooting zone depth in section 2.2 (page 4, line 8 – line 11), when we give more details about the land covers investigated in this study: "The soil root zone in the wooded Cerrado may reach up to 18 m (Rawitscher, 1948). However, most of the water used for plants' transpiration comes from the first layers (up to 7.5 m) (Canadell et al., 1996; Oliveira et al., 2005; Garcia-Montiel et al., 2008)."

**R1C8: Statistical data analysis was done well with good uncertainties estimations.**

AR-R1C8: We appreciate this comment.

**R1C9: Chapter 3. With the tables and figures the results are consistent documented and described. Discuss more on page 8, row 20-25: why in table 5 results for Eta differed (because of different sites with different rainfall amounts, because of different methods e.g. Nobrega.**

AR-R1C9: This is an important remark. We appreciated the recognition of our Tables and Figures consistency. We noted significant different values among the listed studies due to the multiple locations and methods considered. Our main idea was to evidence this huge variability among the reference studies. Thus, we added an extra piece of discussion on page 9 (line 7 – line 8) to clarify the main reason for these differences and change the paragraph structure. "(…) (Table 4) due to the diverse

rainfall patterns among the study sites and the different methods used to measure or estimate the evapotranspiration."

**R1C10: Page 8 row 30 following: discuss more the uncertainty of Cerrado vegetation rooting zone for the evapotranspiration calculation (depth of rooting zone you used is very sensitive for the residual in the water balance.**

AR-R1C10: Thank you for this important remark. Actually, the methods used for the evapotranspiration estimates in the wooded Cerrado did not use the root depth as input parameter (see Equation 7). However, we recognize that it is important to discuss about the root zone depth in this paragraph. Thus, we added a brief discussion stating about the root zone uncertainty in the Cerrado (page 9, line 18 – line 21). "However, the root zone depth of an undisturbed vegetation such as the wooded Cerrado is uncertain and may vary according to the soil characteristics (Canadell et al., 1996) and water table level (Leite et al., 2018). It may influence the plants' transpiration (Rawitscher, 1948; Oliveira et al., 2005), and consequently the water balance residual."

**R1C11: Results for LUC to pasture are well in accordance to other studies, role of soil compaction should be discussed for this land use (see Nobrega 2017 and Meister et al. 2017).**

AR-R1C11: Yes, it is true to say that soil compaction in pasturelands will affect the water balance results. Thus, we changed the text at this point (page 10, line 8 – line 10) to consider the suggested studies, in order to enrich our discussion: "Additionally, the deforestation and agricultural land uses may increase soil compaction, as the LCLUC influence the hydrological patterns along the soil profile by evident modifications in the soil characteristics (bulk density, infiltration capacity, etc.) (Lamparter et al., 2016; Meister et al., 2017; de Almeida et al., 2018)."

**R1C12: Page 9 row 25 on: the chapter is misunderstanding comparing with Fig 4 (water table changes): Row 26: water balance residuals represent not only soil water storage, as defined before (includes also deep infiltration – groundwater recharge !); authors argues that cerrado remove water from deeper soil horizons (that's right), but groundwater fluctuation is much higher in pasture and sugarcane (why?.**

AR-R1C12: Thank you for this important question. We explained this issue on page 11 (line 8 – line 18, and line 21 – line 24). This happens because more water reaches the water table in the pasture in comparison with the Wooded Cerrado. In the wooded Cerrado, the water uptake by the vegetation is higher due to the deeper and denser root system in comparison with pasture and sugarcane. In the sugarcane and pasture, the soil water that was not consumed by the plants and neither evaporated, it continues to infiltrates along the unsaturated zone and the water uptake by the plants becomes unfeasible as the roots are shallow. Consequently, more water becomes available for deep infiltration, and this is evidenced by the significant water table fluctuation, which means that maybe there is a higher groundwater recharge under the pasture in comparison with the wooded Cerrado (Fig 3f). This is

explained in detail in section 3.2. Sections 3.1. and 3.2. had their contents changed due to the new information added in Fig. 3.

**R1C13: It will be fine, if table or figure with the soil water content over the measurement period can be added, than it can be seen how the unsaturated soil zone react different between land uses and cerrado. – In Fig. 4b 2015 there is a remarkable water table deepening, but high surplus of dS/dt – why?**

AR-R1C13: We inserted a Figure showing the soil water content along the monitoring period and we attached in this revision as Fig. 3e. To answer the question about the water table deepening in 2014, we added a piece of discussion in the text (page 10, line 32 – line 33): "In the well located in the pasture, the water table fluctuated negatively along 2014 and 2015 due to the drought that happened in 2014 (Getirana, 2015). The water surplus of 2015-2016 happened due to the La Niña phenomena, that raised the rainfall pattern after the long dry season of 2014-2015 (Kakatkar et al., 2018). Consequently, the water table raised along 2016-2017."

**R1C14: Discussion chapter 3.4 (should be enlarged a little with): Result that pasture and sugarcane increase surface runoff and decrease Eta are very common (not surprising); but for the residual (increased significantly) it must be discussed more carefull with differentiation in the role of deep infiltration (groundwater recharge relative high, interflow in the slope?, change of soil water content – see the measurements – not used for the discussion; infiltration rates between Cerrado and land use types are comparable? Compare with literature results.**

AR-R1C14: Thank you for the suggestion. We added an explanation and new citations (a number of 8) to explain why the groundwater fluctuation is higher in the pasture well than in the Cerrado: "Therefore, the aquifer recharge rates, evidenced here by the groundwater table fluctuation (Fig. 3F), may be reduced in forested areas in comparison with agricultural landscapes due to the atmospheric and vegetation water demands, and the increased soil water retention capacity (Adane et al., 2018; Dias et al., 2015; Wang et al., 2018; Tseng et al., 2018). This validate the information that the LCLU significantly impacts groundwater recharge (Scanlon et al., 2005; Scott et al., 2014; Lucas et al., 2015; Dawes et al., 2012)" (page 13, line 11 – line 15). In addition, phrases on lines 3, 4, 26 and 27 (page 13) were added to complete the discussion. This issue was mainly discussed along the section 3.3, which was significantly improved in the revised version of the manuscript. The hole of section 3.4 is to evidence the hydrological trade-offs caused by the potential LCLUC explored by our study.

**R1C15: Conclusions: page 12, row8: avoid term change in soil water storage (you mean the residual, much more than soil water storage (see above) Page 12 row 11,12: no documentation that higher infiltration rates in wooded Cerrado compared to pasture and sugar cane – add this in the paper.**

AR-R1C15: We changed the term "soil water storage" through the text to "water balance residual" and we defined this terminology on session 2.3 (page 4, line 15 – line 16). "The water balance residual

(dS/dt) includes subsurface flow, soil water storage, deep percolation and groundwater recharge." We commented along the text that due to the decreased runoff of the wooded Cerrado in comparison with pasture, sugarcane and bare soil, more water infiltrates through the soil and become readily available for the plants' consumption (page 10, line 14 – line 15): "These reduced surface runoff rates in the wooded Cerrado increase soil water infiltration in comparison with pasture, sugarcane and bare soil. Thus, higher infiltration rates increase plant water availability (Krishnaswamy et al., 2013)."

**R1C16: I agree, that such long term monitoring studies must be done, to compare it with often done pure water balance simulation studies. Point out in 4., what for important results in detail are valuable for further studies and water balance modelling for Cerrado Biome. In total: acceptance with mayor revision**

AR-R1C16: We acknowledge the insightful comments about our manuscript and the kind words in support of its publication. We added in conclusions that our results are useful for future research, both for discovery and modeling sciences.

**R1C17: Please add in the references: PROCESS-BASED MODELLING OF THE IMPACTS OF LAND USE CHANGE ON THE WATER BALANCE IN THE CERRADO BIOME (RIO DAS MORTES, BRAZIL) Sarina Meister, Rodolfo L. B. Nobrega, Wolfgang Rieger, Ronja Wolf and Gerhard Gerold ERDKUNDE 2017, 71/3, 241-266 Lamparter, G.; Nóbrega, R.. L. B.; Kovacs, K.; Amorim, R. S. and Gerold, G. (2016): Modelling hydrological impacts of agricultural expansion in two macro-catchments in Southern Amazonia, Brazil. In: Regional Environmental Change. https://doi.org/10.1007/s10113- 016-1015-2**

AR-R1C17: Thank you for the suggested references. As commented in AR-R1C11, we cited these studies to support the discussion of our results.

Reviewer 2 comments

We would like to thank the anonymous referee 2 for the kind words in support of our manuscript and for the time spent reviewing our text. Here, we replied the referee's comments, which were highly insightful and enabled us to improve the quality of our manuscript. Note that the original referee's comments are identified as R2Cxx and written in **bold**, and the authors' responses are labeled as AR-R2Cxx. In addition, all comments are numbered (xx).

**R2C1: The authors assess the impact of four different land uses (bare soil, sugarcane, pasture and wooded cerrado) on water balance components (mainly runoff and actual evapotranspiration) monitored in experimental plots (5 m width, 20 m length and 9% slope). The paper is potentially interesting for the readers of Hess, however it requires major revisions.**

AR-R2C1: We appreciate the reviewer's comments and suggestions and we recognize that our manuscript requires more detailed descriptions of some topics listed by Reviewer 2. We hope to solve the problems found along the text to improve its comprehension and quality. In addition, it is important to remark that this is a site-specific study, and the discussion and conclusions stated in here are based on field observed data, without any pretention to generalize to a larger area. Our scope was to investigate the hydrological process in detail by adopting the hillslope scale as the design criteria of the experimental setup. Thus, our results will help further studies supporting them with 5-year experimental data that confirms the significant influence of the land use in the water partitioning in a subtropical region.

**R2C2: In sub-Section 2.2 ("Experimental setting and instrumentation"), please clearly describe the monitoring infrastructure (refer to Fig. 2) by adding information on soil moisture probes, monitoring wells which are described in other parts of the manuscript. Please add the thickness of the unsaturated zone (40 m) in Fig. 2.**

AR-R2C2: Thank you for the suggestion. We added the thickness of the unsaturated zone in Fig. 2 and additional information was added in section 2.2 (page 3, line 17 – line 25) to better describe the monitoring wells and soil moisture probes.

**R2C3: Not clear Kc-values for sugarcane in Table 2. Do they refer to monthly values during the growing season? How do you obtain field capacity and saturated hydraulic conductivity? Please specify the root zone depth for sugarcane and wooded cerrado.**

AR-R2C3: Thank you for requesting this information. The Kc value vary along the year. We specified in a new version of Table 2 the months that the different Kc values are referring to. We obtained the soil field capacity and saturated hydraulic conductivity using Büchner funnels and Richards extraction chambers (information to be added in the revised manuscript). The root zone depth for sugarcane was specified on Table 2 (see the second column). However, the root zone depth for the wooded Cerrado was specified in the revised version in section 2.2 (page 4, line 8 – line 11): "The soil root zone in the

wooded Cerrado may reach up to 18 m (Rawitscher, 1948). However, most of the water used for plants' transpiration comes from the first layers (up to 7.5 m) (Oliveira et al., 2005; Canadell et al., 1996)".

**R2C4: In sub-Section 3.2 ("Groundwater table fluctuation"), the results are suspicious. The authors declare Ks=10-3 m d-1 for a sandy soil (very low if compared to tabulated values, see publications of Clapp and Hornberger, 1978; Schaap et al., 2001; Twarakavi et al., 2009 to mention a few). The main problem is the relationship between water storage change and water table fluctuation in Fig. 4. If Ks=10-3 m d-1, and hypothesizing full saturation of soil profile, we can apply the Darcy law with the unit gradient and water takes about 4000 days to bypass 40 m of soil and reach the water table. Please check if I am wrong. If I am approximately right, the relationship between soil water storage change and water table fluctuation should be influenced by a time lag.**

AR-R2C4: Thank you for the remark. Our idea was to verify the aquifer hydraulic conductivity in both wells (pasture and wooded Cerrado) to be sure that the wells conditions were the same and consequently comparable in terms of water table fluctuation. The hydraulic conductivity mentioned in the section 3.2 was obtained by a slug test. Therefore, it is not referring to the hydraulic conductivity for the sandy soil from the unsaturated zone. In fact, it is referring to the aquifer hydraulic conductivity, which is a sandstone. The soil hydraulic conductivity in the upper layers of the soil (up to 1 m depth) is around 10-1 m d-1. Thus, the soil porosity, and consequently, the soil hydraulic conductivity may vary along the unsaturated zone. We totally agree that there is a time lag between the water infiltration and the aquifer recharge, thus, we added extra pieces of discussion on section 3.2 (pages 10 and 11) to clarify that the effects observed on the groundwater table fluctuation tend to be non-linear and difficult to analyze as they result from complex interactions between LCLU and hydrological processes (Han et al., 2017). We also believe that the discussion from section 3.2 states a new hypothesis to be tested in further studies, in order to check the time lag between surface water interactions and groundwater recharge in contrasting LCLU (page 11, line 16 – line 18).

**R2C5: The main concern of this study is that the authors draw general conclusions on a small-scale, quite "homogeneous" test-site (experimental plots of 5 m width, 20 m length and 9% slope) by ignoring large-scale spatial heterogeneity of soil properties and topography. Hence this study can be considered as a preliminary survey for a more ambitious scientific investigation.**

AR-R2C5: We appreciate your concern about the scale factor. There is a need to understand the water partitioning in the Cerrado region at multiple scales (Oliveira et al., 2014). In this study, our priority was to investigate the water balance at the hillslope scale in multiple land uses. The choice of the evaluated LCLU was based on the current LCLUC dynamics in the Cerrado described in the introduction and in the comment AR-R1C2. In the future, the findings of the present study can be used for data validation in spatial scale studies and be part of a multi-scale approach to evaluate the water balance in the Cerrado.

Reviewer 3 comments

We would like to thank the anonymous referee 3 for the kind words in support of our manuscript and for the time spent reviewing our text. Here, we replied the referee's comments, which were highly insightful and enabled us to improve the quality of our manuscript. Note that the original referee's comments are identified as R3Cxx and written in **bold**, and the authors' responses are labeled as AR-R3Cxx. In addition, all comments are numbered (xx).

**R3C1: This paper describes an experimental approach at the hillslope scale concerning the possible water partitioning trade-offs due to the LCLUC dynamics. I think this paper is relevant since it studies water flux in the Cerrado biome. The manuscript is interesting and well written. The results are original and represent an important contribution to the understanding of hydrological processes in the Cerrado. However, my main concern is that the problem statement is not clearly defined and that the field experimental description is not sufficient as it is. I think the paper is well written and the relevant literature cited, however it requires major revisions.**

AR-R3C1: We appreciate the reviewer feedback about our manuscript and the time spent during the reading and further revision. Here, we express our accordance with the relevant given suggestions. We also added the information in the revised version of the manuscript. We deeply revised the problem statement, which is mainly based on the lack of field studies in the tropics considering different land uses and an undisturbed condition (Cerrado). We recognize that it could contain more arguments involving the need of field observed data for both modeling and discovery sciences.

**Suggested corrections:**
**R3C2: All acronyms of the equations that are in the text should be in italic and the equation with the unit (equation 8).**

AR-R3C2: Thank you for the correction. We changed this along the text according to the instructions.

**R3C3: Page 3: The figure 1, is it to highlight Brazil? the most important is the monitoring, the details in the photos are too small. Please improve visibility.**

AR-R3C3: Thank you for your suggestion. We wanted to highlight the study area context by printing the country larger than the other information. However, we recognized that it is more important to show the reader more information about the study site itself. Thus, we made a new Fig.1 considering this important suggestion.

**R3C4: Page 3: In sub-Section 2.2 Experimental setting and instrumentation-better describe the part of the monitoring. What was the measuring range for each equipment? What are the distances between the equipments? What is the size of the plot? What are the characteristics of the forest? (DBH, Height, Density)**

AR-R3C4: Thank you for these important questions. Concerning the measuring range of each equipment, we added this information on Table 1. The distance between sites 1 and 2 are 1.7 km. Site 1 has 9 plots placed side by side (approximately 2.5 meters of distance between each plot) (see Figure 1). Also in site 1, there is a meteorological station that concentrates almost all sensors placed at that site at one point, except for the soil moisture probes. They are connected to the meteorological station, but they are placed inside the plots (there is one placed inside the first sugarcane plot, and 20 m to the left, there is another one placed inside the first pasture plot). Site 2 has 3 plots inside a tropical woodland (wooded Cerrado) and due to the tree density and topography, the plots are approximately 5 to 10 meters distant from each other. Approximately 50 m to the north from the plots, there is a meteorological tower (11 m height) containing all the sensors placed at site 2. Concerning the forest characteristics, the wooded Cerrado area used in this study has 15522 individuals per hectare, the height of most of the trees is about 8 m, and the diameters (DBH) are predominantly between 3 and 7 cm (Reys et al., 2013). We added extra information in section 2.2 in order to better describe the instruments' positions (page 3, line 17 – line 25) and the forest characteristics (page 4, line 6 – line 9).

**R3C5: In table 1: Were used different equipment for monitoring the same variable in different plots (i.e. soil moisture)? What is the error of each piece of equipment? some tests were carried out to know the difference between devices?**

AR-R3C5: Thank you for this important remark. We used the same model of soil moisture probes in the different measurement points (see Table 1). We added the maximum error of each piece of equipment in Table 1. The soil moisture probes had their first use in our study sites and they were all previously calibrated with soil samples from our study sites.

**R3C6: In figure 2, Why did not your measure soil moisture in the Bare soil? Please explain.**

AR-R3C6: Thank you for the question. We had not enough ports in the datalogger to connect another soil moisture probe to monitor the bare soil plot. In addition, we did not have a piece of equipment available to perform such monitoring.

**R3C7: Page 4: The paragraphs in lines 5 to 13, should be inserted in the sub-section 2.2.**

AR-R3C7: We agree that this information suits on section 2.2. However, we added these paragraphs in section 2.3 (water balance components) because each of them is describing how we obtained each of the water balance components: Page 4 (line 22 – line 23) describe how we monitor the rainfall; Page 4, line 24 – line 26 describe how we obtained the overland flow and how we calculated the runoff coefficient; Page 4, line 27 – line 29 describe how we estimated the reference evapotranspiration. Thus, we argue that removing these paragraphs from section 2.3, we may lose the sequence of the text. You can see that just after line 16 (page 4), we present Equation 1, which is also referred in line 16. Additionally, these paragraphs define how we obtained the input variables used in Equation. 1.

**R3C8: Page 5: How and when do you obtain soil field capacity and saturated hydraulic conductivity?**

AR-R3C8: We obtained the soil field capacity and soil wilting point using Büchner funnels and Richards extraction chambers (Richards, 1931). These tests were performed in the beginning of the experiment (2012) (Oliveira, 2014). We collected the samples with undisturbed structure in volumetric rings at depths of 20, 50 and 100 cm. These information were added to the text (page 5, line 29 – line 31). The soil hydraulic conductivity range found in the study area was added on page 3 (line 9 – line 10).

**R3C9: Page 5, Table 3: remove this table, you can describe it in a paragraph.**

AR-R3C9: Thank you for the suggestion. We removed Table 3 and a new paragraph was added after line 5 (page 6): "The Priestley and Taylor coefficients (α) calculated for a wooded Cerrado area close to the study site (Cabral et al., 2015) differed according to the season: 1.09 for Summer (December – March); 1.00 for Fall (March – June); 0.77 for Winter (June – September); and 0.98 for Spring (September – December).".

**R3C10: Page 6: In sub-section 2.4 and 2.5 describe more about these topics.**

AR-R3C10: Thank you for the suggestion. We completed these 2 paragraphs with additional information. We modified the paragraphs as reported below:

Section 2.4: Groundwater table fluctuation (page 6, line 24)
"The water table was registered twice a day (at 6 am and 6 pm) using pressure transducers (Diver, Schlumberger) placed inside two monitoring wells (well 1 located in the pasture area; and well 2 located inside the wooded Cerrado area). In the study site, both wells presented similar hydraulic conductivity according to the slug test (Bouwer and Rice, 1976) previously performed. We evaluated the aquifer hydraulic conductivity from both wells in order to validate the water table comparison among each other, as whether the aquifer condition in the wells were different, such comparison would not be fair. Both wells reach the water table at approximately 40 m depth in an unconfined sandstone formation (Botucatu formation), which belongs to São Bento Group of Mesozoic age. In addition, the soils above the aquifer that appears thought the unsaturated zone are Cenozoic sediments weathered from the sandstone (Wendland et al., 2007)."

Section 2.5: Data analysis (page 7, line 7)
"The normality assumption was tested using the Shapiro-Wilk test using a 95% confidence interval for rainfall, evapotranspiration, surface runoff and soil water storage datasets. The one-way analysis of variance (ANOVA) was applied to test the null and alternative hypothesis, that is, equality of surface runoff, evapotranspiration and soil water storage distribution functions between the four treatments (LCLU) versus the difference in distribution functions between at least two treatments. Additionally, the multiple comparisons between treatments were performed using the Tukey test (Montgomery, 2008).

The rainfall, evapotranspiration, surface runoff, water balance residual and soil moisture graphs were plotted using a daily basis timescale. The groundwater table fluctuation was plotted using a monthly timescale due to the noise typically found in this kind of measurement. In order to present the order of magnitude along the years, the data was also resumed annually in Tables and Figures."

**R3C11: Page 8, Results and discussion: I think you need to further describe the results and compare with other papers. The study would have been of more interest to readers if various published water flux models had been tested using the data.**

AR-R3C11: Thank you for your suggestion. As asked by other reviewers too, we improved the results' discussion by contrasting our outcomes with other studies in the revised manuscript to be submitted. This can be evidenced by the 27 new references included in the manuscript. Concerning the testing of water flux models, it was not part of our scope to test models using our data. We consider that water flux model testing with the data presented along this study should be part of a future study. Thus, we may add it as a recommendation along the discussion and also in the concluding remarks. We believe that the main contribution of our study is the long-term monitoring at the hillslope scale under subtropical conditions. Such kind of data is a resource for both discovery and modeling sciences. Additionally, we could draw significant conclusions by the comparisons of the contrasting land uses considered in this study.

**R3C12: Page 12 Conclusions: The conclusion reads more like a summary of the paper.**

AR-R3C12: We recognize this aspect. Along the revision, we added substantial information in the results and discussion session. Therefore, in the revised version, we added other assumptions discussed thought the manuscript. (page 14, line 2 – line 6, and line 14 – line 15), mainly stating the two main reasons of a higher water consumption observed in the wooded Cerrado in comparison with pasture and sugarcane. The fact that we summarize the manuscript in the first paragraph of the conclusion is due to the need of remember the reader about the context of our study. In addition, some readers go straight to the conclusions in a first read of a paper and when we give at least a brief description of the study before giving the conclusions, we improve the comprehension of our scientific contributions. However, we added more information along the conclusions to reinforce the accomplishment of the objectives and complete the main findings.

[revised manuscript text omitted]

(1)

Where: $P$ is the rainfall (mm); $Q$$O_F$ is the surface runoff (mm); $E_T$ is the evapotranspiration (mm); and $dS/dt$ includes the soil water storage, subsurface flow and deep percolation (mm) during time $t$ (day, month or year).

The rainfall was monitored using a tipping bucket (Model TB4-L, Hydrological Services) with a gauging resolution of 0.254 mm (Table 1). The rainfall data was registered every 10 minutes, in order to obtain rainfall intensity and duration.

Rectangular experimental plots directed the overland flow to tanks at the end of the slope, where the volume was measured at times after rainfall events. We also calculated the runoff coefficient for each LCLU using a genetic algorithm to minimize the squared errors between observed and estimated runoff values using the rational method (Wang, 1991).

The reference evapotranspiration ($E_{To}$) was calculated on a daily-basis using the Penman-Monteith equation parameterized by FAO 56 methodology (Allen et al., 1998). Afterwards, the evapotranspiration ($E_T$) values were obtained for pasture and sugarcane land uses using Eq. 2.

$$ET = K_s \cdot ET_c = K_s \cdot (K_c \cdot ET_o) \quad E_T = K_S \cdot E_{Tc} = K_S \cdot (K_C \cdot E_{To})$$

(2)

[revised manuscript text omitted]

Commented [JAAA4]: Removed figure.

[Figure]

**Figure 5: Annual soil water storage**

Commented [JAAA5]: Updated figure, see below.

[Figure]

**Figure 4: Annual water balance residual** ($dS/dt$) **for different LCLU during 2012-2016 period; shaded areas indicate the standard error (uncertainties) of** $dS/dt$ **estimates.**

[Figure]

Commented [JAAA6]: Updated figure, see below.

[Figure]

**Figure 65: Annual observed means (2012-2016) of hydrological trade-offs related to evapotranspiration ($E_T$), surface runoff (Q), soil$O_F$), water storagebalance residual (d$S$/d$t$) due to potential LCLUC found in southeastern Brazil.**

---

## Author Response (AR2)

To: Dr Martijn Westhoff
Associate Editor
*HESS*

From: Dr Jamil A. A. Anache
Corresponding author
*EESC-USP*

Re: Manuscript reference HESS-2018-415

February 12th, 2019

Dear Dr. Westhoff,

We would like to thank you for considering our manuscript entitled "**Hydrological trade-offs due to different land covers and land uses in the Brazilian Cerrado**" by Jamil A. A. Anache, Lívia M. P. Rosalem, Edson Wendland, Cristian Youlton and Paulo T.S. Oliveira for publication in Hydrology and Earth System Sciences. In the final uploaded manuscript, we corrected a mistake in Eq. 11 and we replaced the minus sign by a multiplication sign, as indicated in the final response letter.

We would like to thank the editor and reviewers for their kind words in support of our manuscript and for their time spent reviewing our text. The editor and reviewers' suggestions were highly insightful and enabled us to improve the quality of our manuscript.

Thank you again for all the support given during the reviewing and discussion processes.

Yours sincerely,

Jamil A. A. Anache
*Post-doctoral research fellow at the Computational Hydraulics Laboratory*
*São Carlos School of Engineering, University of São Paulo*